# Genetic mapping of etiologic brain cell types for obesity

**Pascal N Timshel, Jonatan J Thompson, Tune H Pers[†]***

Novo Nordisk Foundation Center for Basic Metabolic Research, University of Copenhagen, Copenhagen, Denmark

**Abstract** The underlying cell types mediating predisposition to obesity remain largely obscure. Here, we integrated recently published single-cell RNA-sequencing (scRNA-seq) data from 727 peripheral and nervous system cell types spanning 17 mouse organs with body mass index (BMI) genome-wide association study (GWAS) data from >457,000 individuals. Developing a novel strategy for integrating scRNA-seq data with GWAS data, we identified 26, exclusively neuronal, cell types from the hypothalamus, subthalamus, midbrain, hippocampus, thalamus, cortex, pons, medulla, pallidum that were significantly enriched for BMI heritability ($p < 1.6 \times 10^{-4}$). Using genes harboring coding mutations associated with obesity, we replicated midbrain cell types from the anterior pretectal nucleus and periaqueductal gray ($p < 1.2 \times 10^{-4}$). Together, our results suggest that brain nuclei regulating integration of sensory stimuli, learning and memory are likely to play a key role in obesity and provide testable hypotheses for mechanistic follow-up studies.

**\*For correspondence:**
tune.pers@sund.ku.dk

**Present address:** [†]Novo Nordisk Foundation Center for Basic Metabolic Research, Faculty of Health and Medical Sciences, University of Copenhagen, Copenhagen, Denmark

**Competing interests:** The authors declare that no competing interests exist.

## Introduction

Identification of genes and cell types underlying susceptibility to human obesity remains a critically important step toward a better understanding of mechanisms causing the disease (*Hekselman and Yeger-Lotem, 2020*). Studies of monogenic obesity syndromes and rodent models of obesity have identified melanocortin signaling circuits in the mediobasal and paraventricular hypothalamus as key components in energy homeostasis and obesity (*Morton et al., 2014*; *Farooqi and O'Rahilly, 2006*; *Betley et al., 2013*). Yet growing evidence suggests that susceptibility to obesity is distributed across numerous brain areas that receive signals emanating from internal sources (e.g. viscerosensory input from the gastrointestinal tract) or external stimuli (e.g. the sight or smell of food) that act in concert to regulate feeding behavior and energy stores (*Grill, 2006*; *Zeltser, 2018*; *Grill and Hayes, 2012*). However, despite an increasing number of genes, cell types and neuronal circuits being implicated in murine energy homeostasis, the identity of brain cell types that drive susceptibility to human obesity remains largely unknown and a systematic assessment of cell types' relevance in obesity is currently lacking.

In recent years, genome-wide association studies (GWAS) have identified about a thousand common (minor allele frequency, MAF $\geq 0.1$) single-nucleotide polymorphisms (SNPs) that associate with body mass index (BMI, defined as weight in kilogram divided by height in meters squared), a heritable and commonly used proxy phenotype for obesity (*Locke et al., 2015*; *Yengo et al., 2018*). In general, the far majority of trait-associated SNPs are located in regulatory regions and hence, unlike coding variants, tagging genetic intervals (or *loci*) rather than implicating specific genes. Importantly, these loci represent an unbiased set of biological sign posts to genes and biological mechanisms underlying susceptibility to obesity (*Hirschhorn, 2009*).

Genetic variants with rare frequencies (MAF $< 0.1$) that are typically too low to be captured in GWAS are thought to contribute ~50% to the heritability of BMI (*Wainschtein et al., 2019*). Many such variants are coding mutations (*Wainschtein et al., 2019*) and hence well-suited to identify causal genes underlying obesity. Lately, rare variant association studies have identified 14 coding

variants across 13 genes in an exome chip analysis across >750,000 individuals (*Turcot et al., 2018*). Interestingly, these genes, with the exception of *MC4R, KSR2* and *GIPR,* have not previously been implicated in obesity, suggesting that key biologic mechanisms underlying obesity have yet to be identified.

Given that a majority of obesity-associated gene variants likely regulate gene expression rather than impact protein function, gene expression data provide an effective scaffold to inform GWAS data for obesity and other traits (*Finucane et al., 2015*; *Calderon et al., 2017*; *Pers et al., 2015*; *Hao et al., 2018*). In 2016, we used microarray-based gene expression data to show that genes in BMI GWAS loci are predominantly expressed in the brain (*Locke et al., 2015*) and we recently leveraged mouse; single-cell RNA-sequencing (scRNA-seq) to implicate mediobasal hypothalamic cell types in obesity (*Campbell et al., 2017*). The growing number of BMI GWAS loci and genes implicated through rare-variant association studies of common and syndromic forms of obesity, in conjunction with the growing number of large-scale scRNA-seq atlases, provide a unique opportunity to systematically uncover genes and cell types underlying biological circuits regulating susceptibility to human obesity.

Here, we developed two computational toolkits for human genetics-driven identification of cell types underlying disease and leveraged them to systematically identify cell types enriching for obesity susceptibility by combining publicly available BMI GWAS summary statistics from >457,000 individuals with scRNA-seq data spanning 380 cell types representing adult mouse organs especially the nervous system and 347 cell types from the adult mouse hypothalamus.

## Results

### Devising a robust cell type expression specificity metric and prioritization framework

Similar to previous approaches (*Campbell et al., 2017*; *Skene et al., 2018*; *Watanabe et al., 2019*; *Bryois et al., 2020*), we hypothesized that cell types exhibiting detectable expression of genes colocalizing with BMI GWAS loci are more likely to underlie obesity than cell types in which these genes are not expressed. Based on this reasoning, we developed CELLECT (**CELL** type **E**xpression-specific integration for **C**omplex **T**raits) and CELLEX (**CELL** type **EX**pression-specificity), two toolkits for genetic identification of likely etiologic cell types. Given GWAS summary statistics and scRNA-seq data, CELLECT can quantify the enrichment of heritability in or near genes specifically expressed in a given cell type using established genetic prioritization models, such as S-LDSC (*Finucane et al., 2015*), RolyPoly (*Calderon et al., 2017*), DEPICT (*Pers et al., 2015*) or MAGMA covariate analysis (*Skene et al., 2018*) (Materials and methods; *Figure 1a*). Importantly, whereas previous frameworks for genetic prioritization of cell types have either relied on non-polygenic models (*Campbell et al., 2017*), used binary or discrete representations of cell type expression (*Finucane et al., 2015*; *Skene et al., 2018*) or used average expression profiles (*Watanabe et al., 2019*), CELLECT uses a robust continuous representation of cell type expression. In Appendix 1, we provide a discussion of our model, its assumptions and relationship to the 'omnigenic' model hypothesis (*Liu et al., 2019*; *Boyle et al., 2017*). Conjointly, CELLEX was built on the observation that different measures of gene expression specificity (ES) provide complementary information and it therefore combines four ES metrics (see Materials and methods) into a single measure ($ES_\mu$) representing the score that a gene is specifically expressed in the given cell type (Materials and methods; *Figure 1b*). We first tested and validated the ES approach on the Tabula Muris dataset (*Tabula Muris Consortium et al., 2018*), a Smart-Seq2 scRNA-seq dataset derived from 17 organs from adult male and female mice, and on the Mouse Nervous System dataset (*Zeisel et al., 2018*), a droplet-based scRNA-seq dataset derived from 19 central and peripheral nervous system regions from late-postnatal male and female mice. For both datasets, we computed gene expression specificities for the four metrics and combined them into $ES_\mu$ across four cell types with known marker genes and found that $ES_\mu$ correctly identified them as being among the most specifically expressed genes (*Figure 1d,e*). We respectively identified a median of 2810 and 4020 specifically expressed genes per cell type and hierarchical clustering of cell types based on the $ES_\mu$ estimates largely reproduced the cell type dendrograms from the respective original publications (*Tabula Muris Consortium et al., 2018*; *Zeisel et al., 2018*), confirming that our ES approach enables cell types profiles to be compared across studies

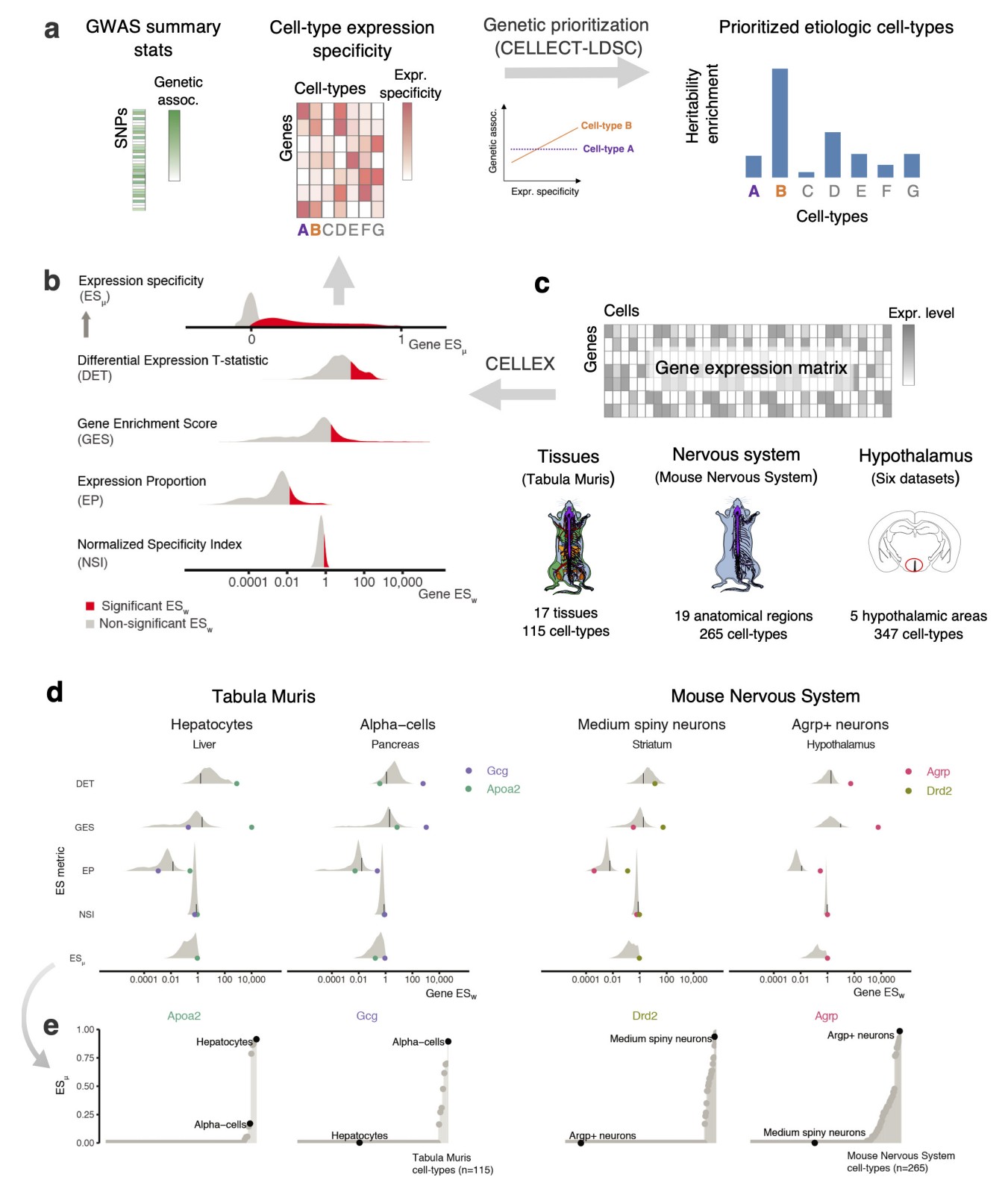

**Figure 1.** Overview of CELLECT and CELLEX and main datasets used (**a**) CELLECT quantifies the association between common polygenetic GWAS signal (heritability) and cell type expression specificity (ES) to prioritize relevant etiological cell types. As input to CELLECT, we used BMI GWAS summary statistics derived from analysis of UK Biobank data (N > 457,000 individuals) and ES was calculated using CELLEX. (**b**) CELLEX uses a 'wisdom of the crowd' approach by averaging multiple ES metrics into $ES_\mu$, a robust ES measure that captures multiple aspects of expression specificity. Prior to

*Figure 1 continued on next page*

*Figure 1 continued*

averaging ES metrics, CELLEX determines the significance of individual ES metric estimates ($ES_w$), indicated by the red and gray colored areas. (**c**) scRNA-seq datasets analyzed in this study. In total, the associations between 727 cell types and BMI heritability were analyzed. Anatograms modified from gganatogram (*Maag, 2018*). (**d**) Example of the CELLEX approach for selected cell types and relevant marker genes. The log-scale distribution plot of $ES_w$ illustrate differences of ES metrics. For each ES metric distribution, a black line is shown to indicate the cut-off value for $ES_w$ significance. In most cases, the ES metrics identified the relevant marker gene as having a significant $ES_w$. In all cases, the marker gene was correctly estimated as having $ES_\mu \sim 1$. We note that the majority of genes have $ES_\mu = 0$ and were omitted from the log-scale plot. (**e**) $ES_\mu$ plots showing the specificity and sensitivity of our approach. The plots depict $ES_\mu$ for the genes shown in panel (**d**) across all cell types in the respective datasets. For each marker gene, the relevant cell type has the highest $ES_\mu$ estimate (high sensitivity) and cell types in which the given gene is likely to have a lesser role have near zero $ES_\mu$ estimates (high specificity). BMI, body mass index; ES, expression specificity; GWAS, genome-wide association study; UK, United Kingdom; scRNA-seq, single-cell RNA-sequencing.

The online version of this article includes the following figure supplement(s) for figure 1:

**Figure supplement 1.** Number of ES genes.
**Figure supplement 2.** Hierarchical clustering of cell types using $ES_\mu$.

and single-cell protocols (*Figure 1—figure supplements 1* and *2*). In Appendix 2, we provide a detailed description of the CELLEX workflow, its assumptions and we use re-sampling to demonstrate the robustness of $ES_\mu$ compared to individual ES metrics. We implemented and released CELLECT and CELLEX as open-source Python packages (see URLs). Here, we – due to its polygenic nature and well-controlled type I error rate – used CELLECT with S-LDSC as the genetic prioritization model to quantify the effects of cell type ES on BMI heritability. For each cell type, we reported the *P*-value for the one-tailed test for positive contribution of the cell type ES to trait heritability (conditional on a 'baseline model' that accounted for the non-random distribution of heritability across the genome, see Materials and methods).

## BMI variants enrich for central nervous system rather than peripheral cell types

Using BMI GWAS summary statistics from a GWAS analysis of the UK Biobank (*Bycroft et al., 2018*) comprising >457,000 individuals (*Loh et al., 2018*) and the Tabula Muris cell types, we first assessed whether we could replicate the exclusive enrichment of BMI GWAS variants in brain tissues as reported by *Locke et al., 2015*. Applying CELLECT to the 115 – mostly peripheral – cell types, we identified two significantly enriched cell types, namely neurons and oligodendrocyte precursor cells (Bonferroni correction-based false-discovery rate, FDR < 0.05; *Figure 2a*). When rerunning CELLECT conditioning on the neuron cell type, the oligodendrocyte precursor cell type was no longer significant, suggesting that we primarily observed a neuronal signal for the BMI GWAS variants. In order to verify that our approach, in general, could identify relevant cell types for complex traits, we computed enrichments for nine GWAS including cognitive, psychiatric, neurological, immunological, lipid and anthropometric traits and disorders, and found that CELLECT prioritized etiologically relevant cell types across all six categories (*Figure 2b*). Cortical neurons were prioritized for cognitive traits and psychiatric disorders (educational attainment, intelligence, schizophrenia), neuronal cell types for insomnia, immune cells for multiple sclerosis and rheumatoid arthritis, growth-related cell types for waist-to-hip ratio (adjusted for BMI) and height, and hepatocytes for low-density lipoprotein levels (see *Figure 2—source data 3* for results across additional 29 traits). Finally, using 1000 'null GWAS' constructed based simulated Gaussian phenotypes with no genetic basis, we found that CELLECT had a properly controlled type I error and that results were not confounded by the median number of genes and transcripts per cell (there was a negligible correlation with the number of cells for a given cell type [Pearson's rho = 0.01, p=$4.0 \times 10^{-4}$], which disappeared when we adjusted for the number of $ES_\mu$ genes for a given cell population [*Figure 2—figure supplement 1*]). These data establish the ability of this approach to validate previous evidence (*Locke et al., 2015*) that BMI variants tend to colocalize with genes specifically expressed in neurons, while also demonstrating that CELLECT is able to prioritize relevant cell types across a number of complex traits.

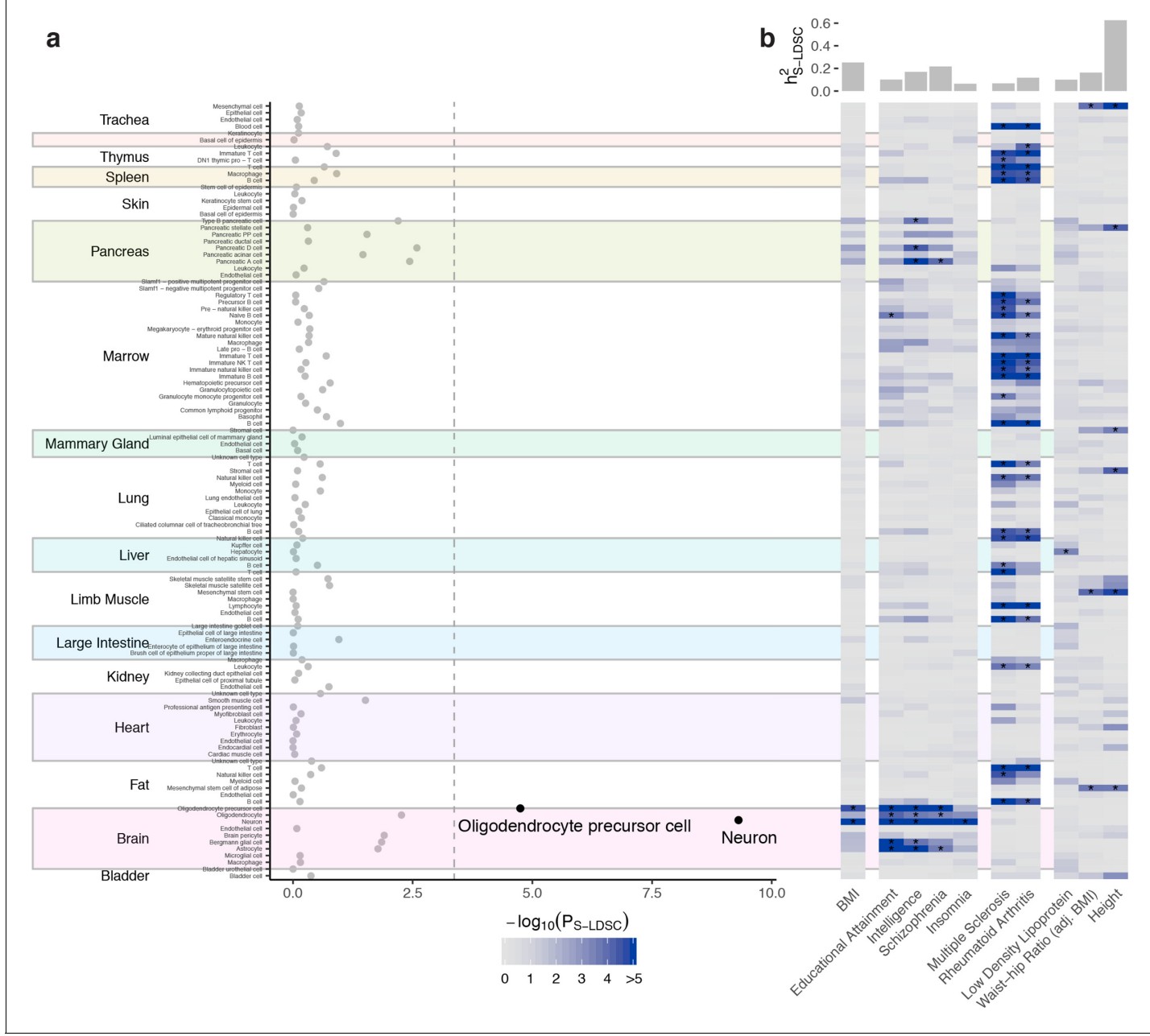

**Figure 2.** Cell type prioritization across 17 tissues highlights a key role of the brain in obesity. (a) Prioritization of 115 Tabula Muris cell types identified two cell types from the brain as significantly associated with BMI, namely oligodendrocyte precursor cells and neurons (shown in black; Bonferroni significance threshold, $P_{S-LDSC} < 0.05/115$). (b) Heatmap of cell type prioritization for multiple GWAS traits. BMI results (first column) are the same as in panel (a) and projected onto the heatmap. The four brain-related traits (second column) were associated with cell types in the brain, the two immune traits (third column) were associated with immune cells, and anthropometric traits (fourth column) were associated with mesenchymal stem cells, which are progenitor cells for muscle, bone and fat. Asterisks (*) mark cell types passing the per-trait Bonferroni significance threshold. The top bar plot shows the estimated trait heritability. An overview of the GWAS files used in this work are available in the *Figure 2—source data 1*, metadata for the *Tabula Muris* dataset are available in *Figure 2—source data 2* and the CELLECT results for the *Tabula Muris* dataset are available in *Figure 2—source data 3*. S-LDSC, stratified-linkage disequilibrium score regression; $h^2_{S-LDSC}$, trait SNP-heritability.

The online version of this article includes the following source data and figure supplement(s) for figure 2:

**Source data 1.** GWAS overview.
**Source data 2.** *Tabula Muris* metadata.
**Source data 3.** *Tabula Muris* CELLECT results.
**Figure supplement 1.** Tests for confounding factors.

## A distributed set of neuronal cell types enrich for obesity susceptibility

We next assessed whether we could identify specific CNS cell types enriching for BMI-associated variants. Applying CELLEX and CELLECT on 265 cell types from the across the Mouse Nervous System dataset, we identified 22 enriched cell types annotated to eight brain regions (*Figure 3a*). To assess the specificity of the BMI GWAS signal in these 22 cell types, we computed enrichments for the panel of nine other well-powered traits. As expected, none of the five traits primarily caused by peripheral etiologies enriched for any nervous system cell type and several of 22 BMI GWAS-enriched cell types also enriched for cognitive traits and psychiatric disorders (*Figure 3b*). Sixteen of the 22 cell types were also enriched 'intelligence' and 'worry', two traits genetically anticorrelated with obesity (overlapping sets of associated loci with opposite effect sizes) (*Marioni et al., 2016*; *Nagel et al., 2018*).

Similar to previous work, we did not find any enrichment of genetic variants associated with BMI in non-neuronal cell types (*Campbell et al., 2017*; *Watanabe et al., 2019*) nor did we detect enrichment for a particular type of neurotransmitter type (*Figure 3—figure supplement 1*). Weighted gene correlation network analysis (WGCNA [*Langfelder and Horvath, 2008*]) on expression data from each of the 22 BMI-enriched cell types identified no significant modules (*Figure 3—figure supplement 2*; top associated module, p=$1.88\times10^{-4}$; FDR $\leq$ 0.1). These findings emphasize that the BMI-associated variants most likely are distributed across hundreds of genes rather than the relatively limited number of genes captured in cell-type-specific WGCNA modules (see Appendix 3 for a discussion on limitations of identifying gene co-expression networks from cell type scRNA-seq data).

To assess the dependence of the results on a given enrichment methodology and BMI GWAS, we re-computed enrichments using the *Yengo et al., 2018* and *Locke et al., 2015* BMI GWAS summary statistics and the MAGMA tool (*de Leeuw et al., 2015*). We observed that the results were robust to different GWAS sample sizes and inclusion of Metabochip array-based association data (Yengo et al. and Locke et al. GWAS Pearson's *R* = 0.98 *and R* = 0.83, respectively), and largely invariant to the enrichment methodology used (Pearson's *R* = 0.82; *Figure 3—figure supplement 3*). Finally, during finalizing this work another study focused on Parkinson's disease, reported BMI GWAS enrichments for the same mouse nervous system cell types (overlap; 6/10) (*Bryois et al., 2020*). Together, these results demonstrate that BMI-associated variants are likely to exert their effect across multiple, predominantly neuronal cell types, several of which enrich for cognitive traits and psychiatric disorders genetically correlated with obesity.

## The enriched neuronal cell types share transcriptional similarities

The 22 BMI GWAS-enriched cell types mapped to eight brain regions, namely the subthalamus, midbrain, hippocampus, thalamus, cortex, pons, medulla and pallidum (*Figure 4a*). To assess the extent to which shared transcriptional signatures could explain the enrichments across the 22 cell types, we clustered all cell types based on their genes' $ES_\mu$ values. Expectedly, midbrain cell types overall grouped by their neuroanatomical proximities and neurotransmitter types by midbrain, hindbrain, hippocampus/cortex clusters (*Figure 4b*). A notable exception was the DEINH3 cell type (isolated from the hypothalamus region and subsequently remapped to the subthalamic nucleus by Zeisel et al.) which grouped with the midbrain cell types. To further assess the transcriptional similarity between the enriched cell types, we computed enrichments conditioned on each prioritized cell type individually (Materials and methods). Contrary to our expectations, we found that none of the other cell types remained significant when conditioning on the top-ranked subthalamic cell type DEINH3 (*Figure 4—figure supplements 1* and *2*). Together these results indicate the brain cell types enriching for BMI GWAS signal, despite their neuroanatomical differences, share transcriptional signatures related to obesity, which current methods are not able to disentangle.

## Ventromedial hypothalamic Sf1- and Cckbr-expressing cells enrich for BMI GWAS

The total number of cell types in the hypothalamus has been significantly underestimated (*Kim et al., 2019a*), therefore to assess whether the lack of enrichment for hypothalamic cell types was due to sparse sampling of hypothalamic cells in the Mouse Nervous System dataset, we computed enrichments for an additional set of 347 cell types sampled from the mediobasal hypothalamus (*Campbell et al., 2017*), the ventromedial hypothalamus (*Kim et al., 2019a*), the lateral

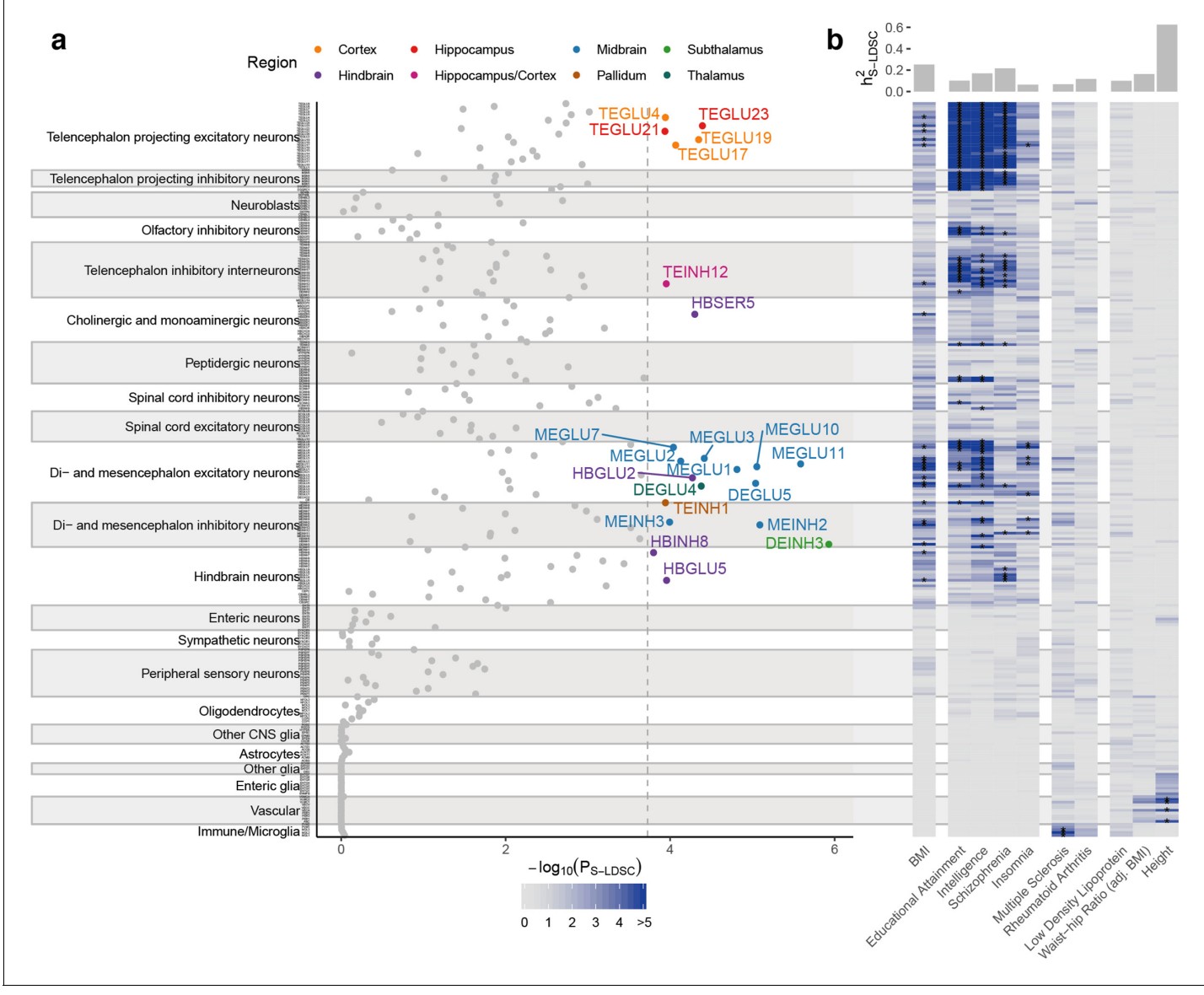

**Figure 3.** Cell type prioritization of mouse nervous system cell types highlights cell types outside canonical energy homeostasis circuits. (a) Prioritization of 265 mouse nervous system cell types identified 22 cell types from eight distinct brain regions as significantly associated with BMI. The highlighted cell types passed the Bonferroni significance threshold, $P_{S-LDSC}$ <0.05/265. Cell types are grouped by the taxonomy described in *Zeisel et al., 2018*. (b) Heatmap of cell type prioritization for multiple GWAS traits. The four brain-related traits (second column) were primarily associated with cortical neurons (telencephalon projecting and interneuron cell types) and did not overlap with the BMI-associated cell types. The two immune traits (third column) were associated with microglia, and anthropometric traits (fourth column) were predominantly associated with vascular cell types. Asterisks (*) mark cell types passing the per-trait Bonferroni significance threshold. The top bar plot shows the estimated trait heritability. Metadata for the *Mouse Nervous System* dataset are available in *Figure 3—source data 1*, CELLECT results for the *Mouse Nervous System* dataset are available in *Figure 3—source data 2*, CELLEX expression specificity values for the BMI GWAS-enriched cell types are available in *Figure 3—source data 3* and cognitive traits and psychiatric disorders CELLECT results limited to the 22 BMI GWAS-enriched cell types are available in *Figure 3—source data 4*.

The online version of this article includes the following source data and figure supplement(s) for figure 3:

**Source data 1.** *Mouse Nervous System* metadata.
**Source data 2.** *Mouse Nervous System* CELLECT results.
**Source data 3.** *Mouse Nervous System* expression specificity results.
**Source data 4.** *Mouse Nervous System* results for other traits and diseases.
**Source data 5.** WGCNA results overview.
**Source data 6.** WGCNA results for the top module M1.
**Source data 7.** MAGMA results.
*Figure 3 continued on next page*

Figure 3 continued

**Figure supplement 1.** BMI-prioritized *Mouse Nervous System* cell type neurotransmitter classes BMI GWAS prioritized cell types prioritization enriched for neurons.

**Figure supplement 2.** Genetic prioritization of cell type gene co-expression networks.

**Figure supplement 3.** Robustness of cell type prioritization results.

hypothalamus (*Mickelsen et al., 2019*), the preoptic nucleus of the hypothalamus (*Moffitt et al., 2018*) and the entire hypothalamus (*Chen et al., 2017*; *Romanov et al., 2017*). We identified four non-overlapping significantly enriched cell populations, namely a ventromedial hypothalamic glutamatergic cell type (ARCME−NEURO29; p=4.9×10$^{-5}$) expressing Sf1 ($ES_\mu$=0.98 and $ES_\mu$=0.99) and Cckbr (cholecystokinin B receptor; $ES_\mu$=0.98, $ES_\mu$=0.95); a glutamatergic cell type from the lateral hypothalamus (LHA-NEURO20; p=4.9×10$^{-5}$); and two cell types from the preoptic area of the hypothalamus (POA-NEURO21 and POA-NEURO66; p<1.0×10$^{-4}$; *Figure 5*). Interestingly, ventromedial hypothalamic neurons have previously been implicated in control of both body fat mass and blood glucose levels; disrupted leptin signaling in Sf1-expressing ventromedial hypothalamic neurons renders mice more susceptible to diet-induced weight gain (*Kim et al., 2011*) and activation of ventromedial hypothalamic Sf1 neurons causes hyperglycemia (*Meek et al., 2016*). The two cell types also expressed Bdnf ($ES_\mu$=0.91, $ES_\mu$=0.99); mutations in *BDNF* and its receptor, *NTRK2*, is a known cause of monogenic obesity in humans and, in mice, BDNF signaling is required for normal energy homeostasis and glucoregulatory control (*Kamitakahara et al., 2016*). (Bdnf and Ntrk2 were also specifically expressed in TEINH12 cell type, a cholecystokinin (Cck)-expressing interneuron, enriched in the mouse nervous system analysis.) Noteworthy, clustering of the 347 hypothalamic cell populations based on their $ES_\mu$ values resulted in clusters predominantly separating by cell type rather than by study or single-cell technique, indicating that CELLEX is relatively robust to batch effects (*Figure 1—figure supplement 2*).

There was no significant enrichment in neurons expressing the Pomc gene, a neuropeptide-encoding gene with known coding mutations causing monogenic obesity in humans (4/5 of Pomc$^+$ cell populations were nominally enriched; HYPR-NEURO24 (Pomc/Ttr), p=0.002; ARCME-NEURO21 (Pomc/Glipr1), p=0.01; HYPC-NEURO23 (Pomc/Cartp), p=0.03; HYPR-NEURO24 (Pomc), p=3.0×10$^{-3}$). We next tested whether the paucity of significant enrichments across hypothalamic populations could be explained by either a limited ability of current hypothalamus scRNA-seq datasets to capture expression of relevant obesity genes or by a limited ability of CELLEX to correctly detect these genes as being specifically expressed in relevant cell types. Towards that end, we first compiled a set of 23 high-confidence obesity genes by merging a set of genes harboring protein-altering variants associated with obesity and with a set of genes implicated in monogenic forms of early-onset extreme obesity (both sets were obtained from *Turcot et al., 2018*; *Figure 5—source data 4*). We then assessed whether these high-confidence obesity genes were robustly- (expressed in ≥10% of the cells in a given population) and specifically ($ES_\mu>0$) expressed within relevant mediobasal hypothalamic arcuate-median eminence complex (Arc-ME) cell populations. By design, Pomc expression was detected in each of the three Pomc$^+$ cell populations; the leptin receptor was detected in agouti-related peptide- and Trh/Cxcl12$^+$ cell populations, two known leptin-sensing cell populations; and, finally, Mc4r was only detected and specifically expressed in the Gpr50$^+$ cell population, which expressed several genes encoding receptors previously related to energy homeostasis (*Campbell et al., 2017*) (Materials and methods; *Figure 5b*, lower panel). CELLEX correctly identified these three genes as specifically expressed in these six cell types. Among the 23 high-confidence obesity genes, 20 were part of the Arc-ME dataset and 17 of them robustly and specifically expressed in at least one neuronal Arc-ME cell population (*Figure 5—figure supplement 1*; *Figure 5—source data 5*). Moreover, four cell populations were enriched for the high-confidence obesity genes ARCME-NEURO21 (Pomc/Glipr1$^+$), ARCME-OTHER1 (a population of non-Arc-ME neurons potentially from the retrochiasmatic area), ARCME-NEURO32 (Slc17a6/Trhr$^+$; neurons shown to be necessary and sufficient to induce satiety [*Fenselau et al., 2017*]) and ARCME-NEURO28 (Qrfp$^+$; an orexigenic neuropeptide involved in energy homeostasis [*Chartrel et al., 2016*]; Bonferroni threshold p<0.05/34; *Figure 5b*, upper panel). We observed a high correlation between the high-confidence obesity gene set- and CELLECT results across the hypothalamus cell

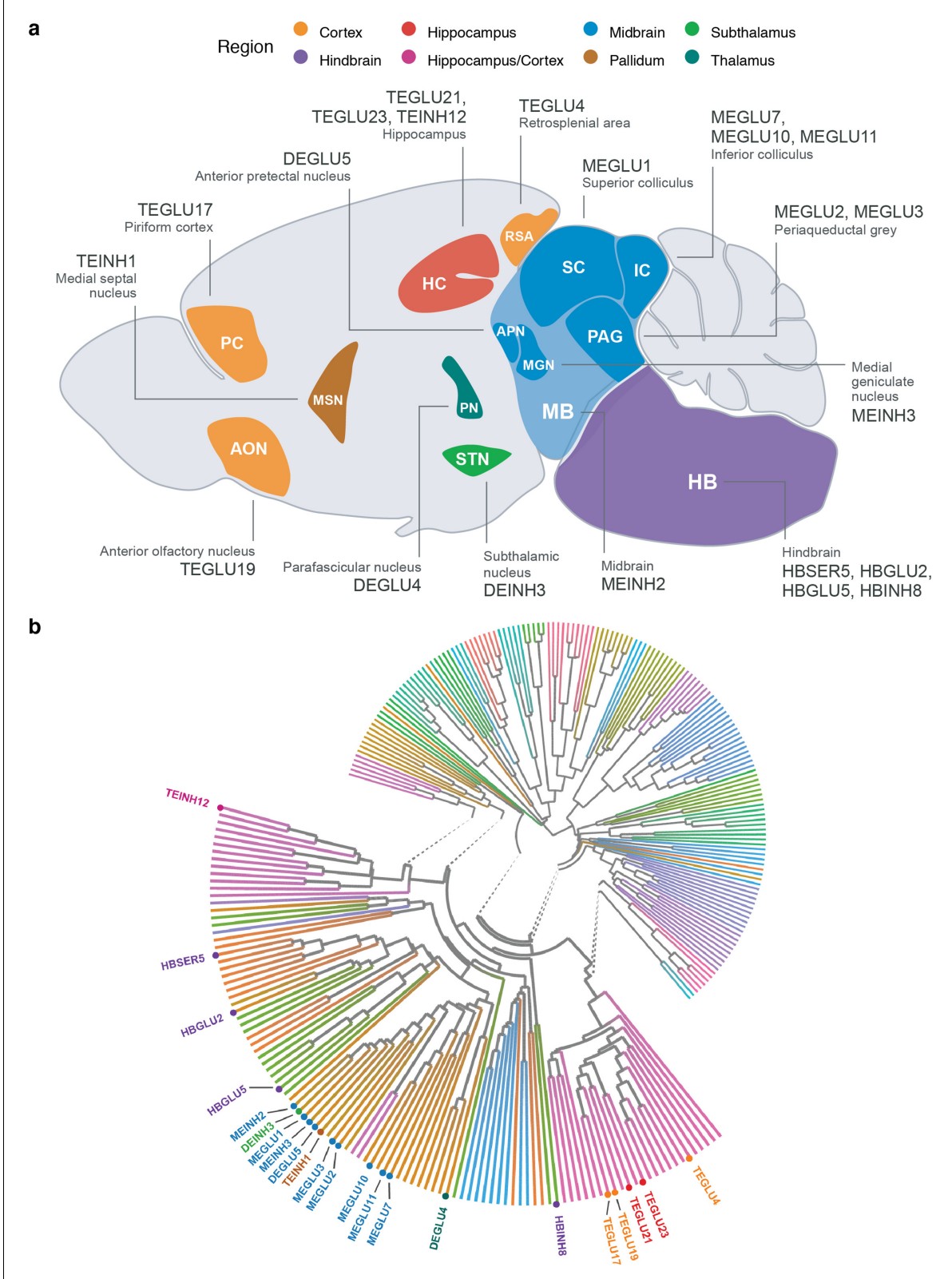

**Figure 4.** Neuroanatomical location and transcriptional similarity of brain cell types enriching for BMI GWAS variants. (**a**) Sagittal mouse brain view showing the 22 BMI GWAS-enriched cell types. The first two letters in each cell type label denote the developmental compartment (ME, mesencephalon; DE, diencephalon; TE, telencephalon), letters three to five denote the neurotransmitter type (INH, inhibitory; GLU, glutamatergic) and the numerical suffix represents an arbitrary number assigned to the given cell type. (**b**) Circular dendrogram showing the similarity of all *Mouse Nervous*

*Figure 4 continued on next page*

*Figure 4 continued*

*System* dataset cell type expression specificity (ES$_\mu$) values. Dendrogram edges colored by taxonomy described in **Zeisel et al., 2018**. Expectedly, the cell types clustered according to their neuroanatomical origin. For clarity, only the labels of the 22 BMI GWAS enriched cell types are shown.

The online version of this article includes the following source data and figure supplement(s) for figure 4:

**Source data 1.** Conditional CELLECT results.
**Figure supplement 1.** Conditional analysis of BMI GWAS-enriched mouse nervous system cell types conditional genetic prioritization analysis of BMI-prioritized cell types.
**Figure supplement 2.** Correlation of mouse nervous system BMI GWAS-enriched cell types correlogram of cell type ES$_\mu$ Pearson's correlations.

types (Pearson's rho = 0.50, p=$1.1\times10^{-5}$; **Figure 5—source data 7**). Moreover, we observed that ES values increased with increasing cell population heterogeneity; 16 out of the 18 ARCME-detected high-confidence obesity genes increased expression specificity when running CELLEX on all Arc-ME cells compared to ARCME neurons-only (**Figure 5—source data 8**). Finally, we found that across the Tabula Muris, Mouse Nervous System and Arc-ME datasets, 22 of the 23 high-confidence obesity genes were among the 25% most specifically expressed genes in at least one cell type (**Figure 5—source data 9**). Together these results indicate (a) that current hypothalamic single-cell data and our CELLEX methodology are of a sufficient quality to detect relevant cell populations, that (b) upcoming regional atlases with increased cellular heterogeneity will drive discovery of additional relevant cell populations and cell states for complex traits, and that (c) the BMI GWAS and high-confidence obesity genes' approaches yield comparable results with a few notable exceptions (such as the Pomc/Glipr1$^+$ population).

Finally, to assess whether hypothalamic transcriptional patterns may explain less genetic heritability compared to other brain areas in humans, we applied CELLEX and CELLECT on RNA-seq data from the Genotype-Tissue Expression Consortium and found that the hippocampus and several other brain areas exhibited stronger genetic enrichment signal than the hypothalamus (**Figure 5c**). In contrast, the high-confidence obesity genes enriched most strongly for the hypothalamus (p=$3.9\times10^{-4}$, FDR < 0.05; **Figure 5—source data 12**). These results support our previous observation that despite overlaps, obesity risk genes identified through rare-variant studies and genes near associated BMI GWAS signals may point to slightly different regions of the brain, an observation highlighting the importance of leveraging polygenic methodologies to identify cell types regulating susceptibility to common obesity.

## Genes with known links to human obesity genes implicate the dorsal midbrain

As the high-confidence obesity genes have been identified independently of the BMI GWAS, we reasoned that we could use them to validate the cell types exhibiting the polygenic BMI GWAS signal. We computed the enrichment of the high-confidence obesity genes within all 265 mouse nervous system cell types and identified eight significantly enriched cell types (one-sided Wilcoxon rank sum test, FDR < 0.05) of which two replicated cell types from the BMI GWAS analysis DEGLU5 and MEGLU2 from the anterior pretectal nucleus and the periaqueductal grey, respectively; p<$1.2\times10^{-4}$. The six remaining cell populations originated from areas implicated by the CELLECT analysis, namely the midbrain (MBDOP1, periaqueductal grey; MBDOP2, ventral tegmental area and substantia nigra; MEINH13, ventral/caudal midbrain; MEGLU14, the dorsal raphe nucleus), the hypothalamus (HYPEP3, ventromedial hypothalamus), and the medulla (HBSER4, nucleus raphe medulla). We observed a significant correlation between the high-confidence obesity genes enrichment- and CELLECT enrichment results (Pearson's R = 0.54, p=$3.0\times10^{-21}$; **Figure 5—figure supplement 2a**), further underscoring the validity of our findings, besides emphasizing that genes implicated in monogenic obesity or implicated by obesity-associated protein-altering variants tend to colocalize with BMI-associated GWAS loci (**Figure 5—figure supplement 2b**; **Locke et al., 2015**).

Interestingly, the leptin receptor, which regulates key energy homeostatic processes in the hypothalamus and when defective may cause syndromic obesity (**Choquet and Meyre, 2011**) was only specifically expressed in two out of the 22 BMI GWAS-enriched in the Mouse Nervous System dataset cell types, namely glutamatergic cells from the periaqueductal grey and anterior nucleus of the solitary tract (**Figure 6a**). By contrast, 17 of the enriched cell types expressed the serotonin receptor

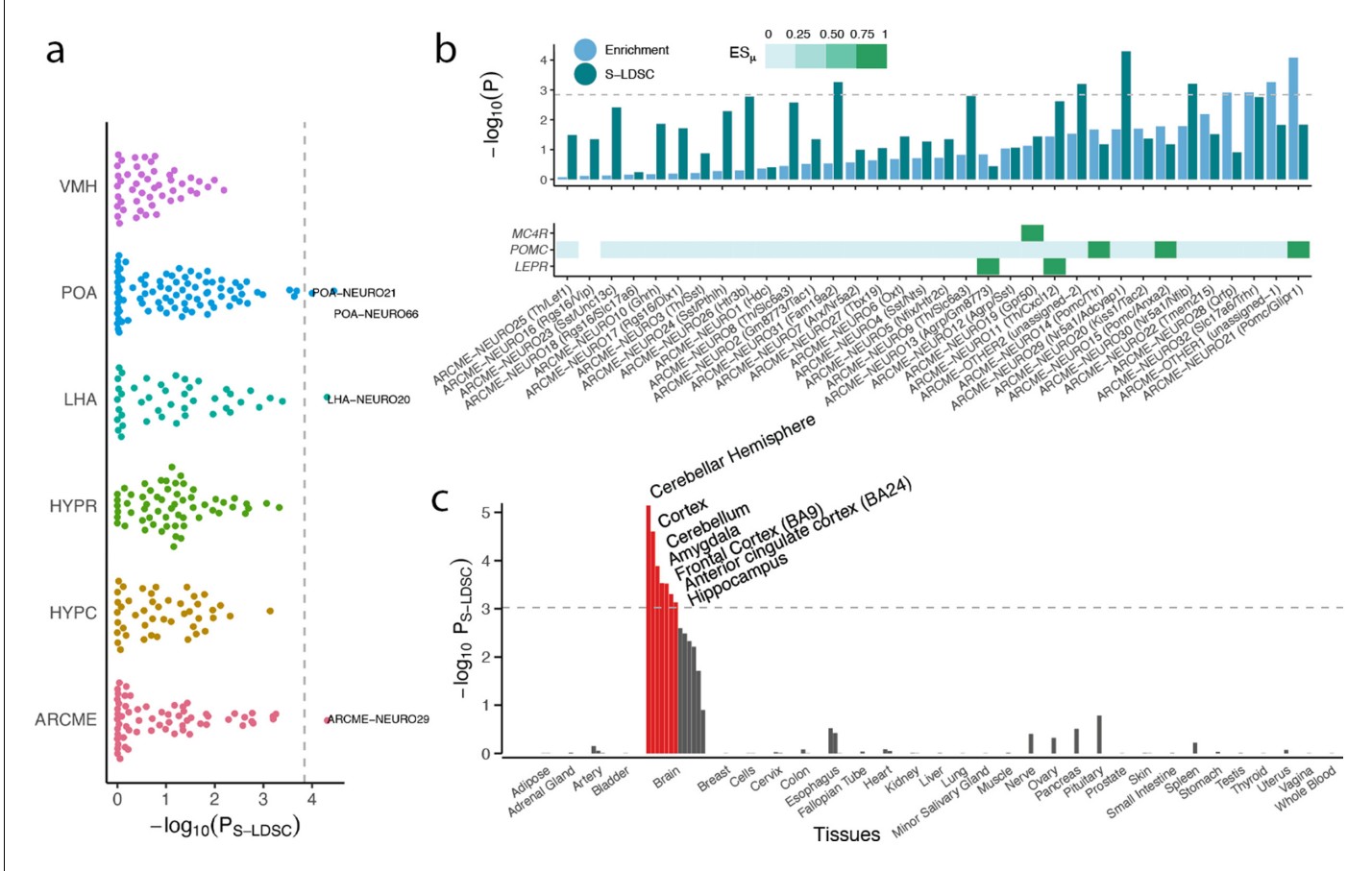

**Figure 5.** BMI GWAS enrichment across hypothalamic cells and human tissues. (**a**) BMI GWAS enrichments across 347 hypothalamic cell types derived from studies of the Arc-ME (ARCME), the ventromedial hypothalamus (VMH), the lateral hypothalamus (LHA), the preoptic nucleus of the hypothalamus (POA) and the entire hypothalamus (HYPR and HYPC). For each study, CELLEX and CELLECT were run individually, and subsequently all cell types were pooled and significance was determine based on Bonferroni correction (p<0.05/347). Four cell types were significantly enriched, namely POA-NEURO66 (Reln[+]; *Moffitt et al., 2018*) and POA-NEURO21 (Cck[+]/Ebf3[+]; *Moffitt et al., 2018*) from the preoptic area of the hypothalamus, ARCME-NEURO29 (Sf1[+]/Adcyap1[+]; *Campbell et al., 2017*) from the Arc-ME, and LHA-NEURO20 (Ebf3/Otb[+]; *Mickelsen et al., 2019*) from the lateral hypothalamus. (**b**) CELLECT and high-confidence obesity genes enrichments for neuronal cell populations in the Arc-ME (upper panel). Expression of Mc4r, Pomc and Lepr across Arc-ME neuronal populations, white squares means that the given gene is not expressed in at least 10% of the cells in the given cell population, non-white squares denote increasingly specific gene expression (lower panel). (**c**) CELLECT enrichment analysis of Genotype-Tissue Expression Consortium (GTEx) RNA-seq data. Orange bars denote significantly enriched tissues. The hypothalamus datasets' metadata, CELLECT results and expression specificity values for the enriched cell types are available in *Figure 5—source datas 1–3*. The GTEx tissue annotations, CELLECT and high-confidence obesity genes enrichment results are available in *Figure 5—source datas 10–12*. POA, preoptic area of the hypothalamus; LHA, lateral hypothalamus; ARCME, arcuate nucleus and median eminence complex; S-LDSC, stratified-linkage disequilibrium score regression.

The online version of this article includes the following source data and figure supplement(s) for figure 5:

**Source data 1.** Hypothalamus datasets metadata.
**Source data 2.** Hypothalamus CELLECT results.
**Source data 3.** Hypothalamus expression specificity results.
**Source data 4.** High-confidence obesity genes.
**Source data 5.** High-confidence obesity genes expression specificities.
**Source data 6.** High-confidence obesity genes enrichments.
**Source data 7.** High-confidence obesity genes CELLECT correlations.
**Source data 8.** Expression specificity and cell type heterogeneity.
**Source data 9.** High-confidence obesity genes CELLEX top quartile.
**Source data 10.** Genotype-Tissue Expression data annotation.
**Source data 11.** Genotype-Tissue Expression CELLECT enrichment results.
**Source data 12.** Genotype-Tissue Expression obesity genes enrichment results.
**Figure supplement 1.** Arc-ME neuronal cell population enrichments and expression levels across obesity genes CELLECT and high-confidence obesity genes enrichments for neuronal cell populations in the Arc-ME (upper panel).

*Figure 5 continued on next page*

**Figure supplement 2.** Convergence of cell type prioritization based on common and rare variants.
**Figure supplement 3.** High-confidence obesity genes enrichment in mouse nervous system cell types.

5-Htr2c (5-hydroxytryptamine receptor 2C), a known regulator of energy and glucose homeostasis (*Berglund et al., 2013*) and a target for anti-obesity pharmacotherapy (*Halford et al., 2011*; *Figure 6b*). 5-Htr2c was most specifically expressed in the anterior pretectal nucleus (DEGLU5, $ES_\mu$=0.96), the cell type among our results which most specifically expressed Pomc ($ES_\mu$=0.41). Mice lacking the Htr2c in Pomc neurons are resistant to 5-Htr2c agonist Lorcaserin-induced weight loss (*Berglund et al., 2013*) (for $ES_\mu$ plots of other selected genes, please refer to *Figure 6—figure supplement 1*).

Together our results indicate that susceptibility to obesity conferred by common variants, while enriching for some hypothalamic cell types such as VMH Sf1-expressing neurons, is distributed across a mosaic of neuronal cell types of which a majority is involved in regulating integration of sensory stimuli, learning and memory.

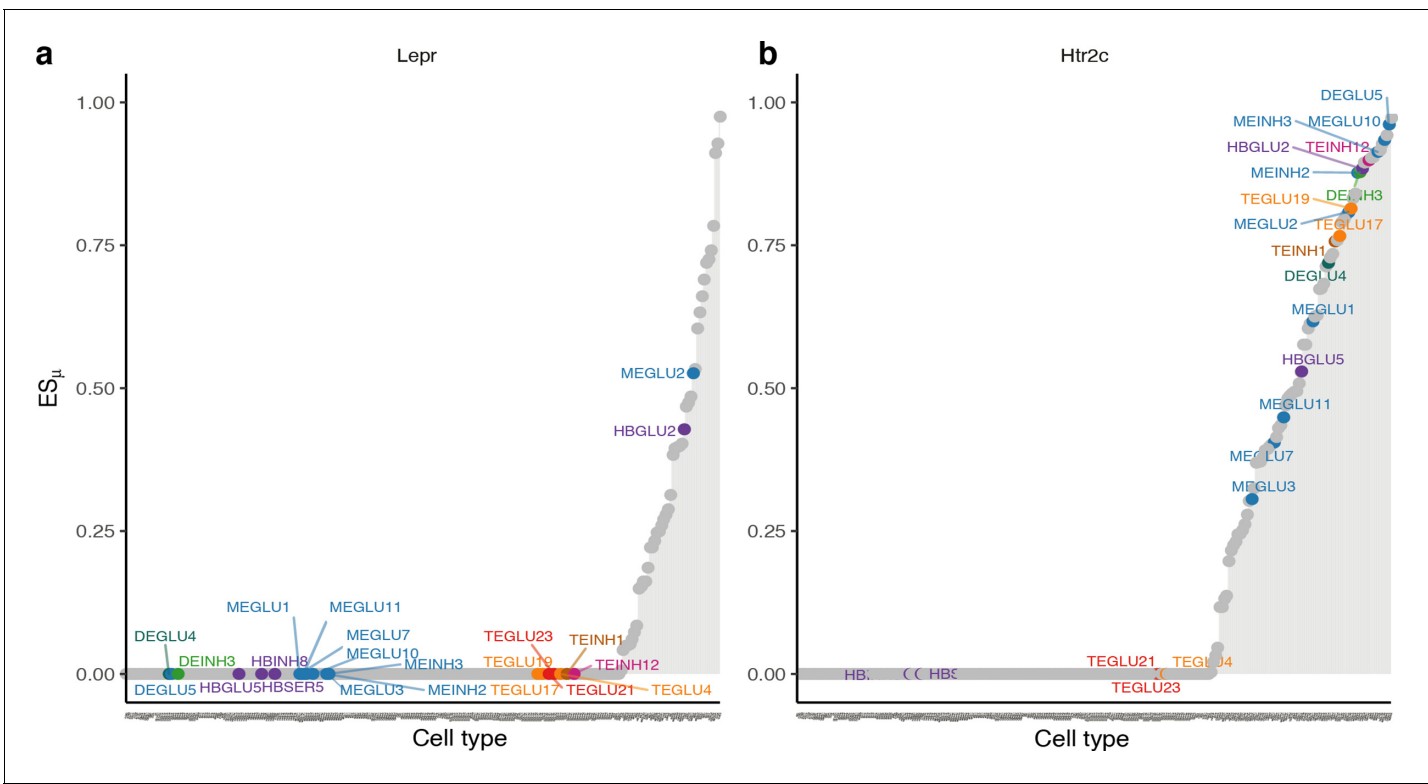

**Figure 6.** Expression specificity of the leptin- and serotonin receptors across BMI GWAS enriched cell types. (a) In the lipostatic model of obesity originally defined by *Kennedy, 1953*, circulating concentrations of the leptin hormone signal the amount of energy stored in fat cells to the brain. The plot shows gene $ES_\mu$ (y-axis) for each cell type (x-axis, ordered by increasing values of expression specificity, $ES_\mu$) with BMI-prioritized cell types from the Mouse Nervous System dataset highlighted. In our analysis, only two of the 22 BMI GWAS enriched cell types specifically expressed the leptin receptor (MEGLU2, periaqueductal grey; and HBGLU2, nucleus of the solitary tract). (b) Seventeen of the 22 BMI GWAS enriched cell types specifically expressed the serotonin (5-htr2c) receptor. The strongest enrichment was observed for DEGLU5, a glutamatergic cell type from the anterior pretectal nucleus. $ES_\mu$, expression specificity.

The online version of this article includes the following figure supplement(s) for figure 6:

**Figure supplement 1.** $ES_\mu$ plot for selected genes $ES_\mu$ plots for genes selected based on their suggested role in appetite regulation, energy homeostasis or obesity.

## Discussion

Here, we developed two scRNA-seq computational toolkits called CELLEX and CELLECT and applied them to scRNA-seq data from a total of 727 mouse cell types, from late postnatal and adult mice, to derive an unbiased map of cell types enriching for human genetic variants associated with obesity. In total, we identified 26 BMI GWAS-enriched neuronal cell types, which, in line with previous considerations (*Grill, 2006*), demonstrates that susceptibility to human obesity is likely to be distributed across multiple, mainly neuronal, cell types across the brain, rather than being restricted to a limited number of canonical energy homeostasis- and reward-related brain areas in the hypothalamus, midbrain and hindbrain. Among the enriched hypothalamic cell types, we identified VMH Sf1- and Cckbr-expressing neurons, which previously have been implicated in glucose and energy homeostasis. We show that while the polygenic enrichment signal is highly correlated with enrichment of high-confidence obesity genes, this alignment diverges for hypothalamic neuron populations (including Pomc-positive neurons) suggesting that common genetic susceptibility to obesity acts on a more broadly distributed set of neuronal circuits across the brain.

### Processing of sensory stimuli and feeding behavior

Several of the enriched cell types localized to nuclei integrating sensory input and directed behavior. The inferior colliculus (implicated by MEGLU7, MEGLU10 and MEGLU11) and medial geniculate nucleus (MEINH3) process auditory input, the superior colliculus for translation of visual input into directed behavior (MEGLU1 and MEGLU6), the anterior pretectal nucleus processes somatosensory input (DEGLU5 and MEINH4), and the piriform cortex (TEGLU17) and anterior olfactory nucleus (TEGLU19) processing odor perception. The superior colliculus and anterior pretectal nucleus have both been implicated in predatory behavior (*Shang et al., 2019*; *Antinucci et al., 2019*) and project to the zona incerta, a less well-described brain area situated between the thalamus and hypothalamus that receives direct input from mediobasal hypothalamic Pomc neurons (*Wei et al., 2018*). In rats, lesioning of the zona incerta impairs feeding responses (*Stamoutsos et al., 1979*) while, conversely, in mouse models, optogenetic stimulation of GABAergic neurons in the zona incerta leads to rapid, binge-like eating and body weight gain (*Zhang and van den Pol, 2017*). Activation of projections from the hypothalamic preoptic nucleus (DEINH5; POA-NEURO21, POA-NEURO66) to the ventral periaqueductal gray (MEGLU2 and MBDOP1) induce object craving (*Park et al., 2018*), whereas pharmacological inactivation of the periaqueductal gray decreases food consumption (*Tryon and Mizumori, 2018*). Together, these findings suggest that susceptibility to obesity is enriched in cell types processing sensory stimuli and directing actions related to feeding behavior and opportunity.

### Evidence supporting a key role of the learning and memory in obesity

Feeding is not an unconditioned response to an energy deficiency but rather reflecting behavior conditioned by learning and experience (*Woods and Begg, 2015*). We previously showed that genes in BMI GWAS loci enrich for genes specifically expressed in hippocampal postmortem gene expression data (*Locke et al., 2015*). In this work, we identified specific brain cell types supporting a role of memory in obesity. First, the parafascicular nucleus (DEGLU4), when lesioned in mice, reduces object recognition memory (*Castiblanco-Piñeros et al., 2011*). Second, the retrosplenial cortex (TEGLU4) is responsible for decisions made on past experiences (*Hattori et al., 2019*). Third, among the two enriched glutamatergic hippocampal cell types (TEGLU21 and TEGLU23), the latter expresses lipoprotein lipase as one of its top marker genes, an enzyme that causes weight gain when pharmacological or genetically attenuated in mice (*Picard et al., 2014*). Similarly, fasting inhibits activation of hippocampal CA3 cells (based on c-fos levels) in mice (*Azevedo et al., 2019*) and activation of glutamatergic hippocampal pyramidal neurons increases future food intake in rats most likely by perturbing memory consolidation related to the previous meal (*Holahan and Routtenberg, 2011*). In sum, our results provide further evidence that processes related to learning and memory play a key role in human obesity, and provide insights into specific cell types underlying hippocampal-centric susceptibility to obesity.

## Limitations of our approach

Our results should be interpreted in the light of the underlying data and methodologies used to prioritize the cell types. First, the scRNA-seq data analyzed here were derived from late postnatal, adult and predominantly wild-type mice; future work is needed to assess the role of Pomc$^+$, Agrp$^+$ and Mc4r$^+$ and other hypothalamic cell types during developmental stages and relevant obesogenic perturbations in human obesity (*Zeltser, 2018*). Second, the datasets used in this work should not be regarded as complete atlases because they are likely to miss relevant cell types such as Mc4r-positive neurons, which are known to play a key role in obesity. Third, one should keep in mind the overall assumption behind our approach, namely that in order for a given gene to confer genetic susceptibility for a given disease it needs to be expressed in the given cell type or tissue, where increasing expression is associated with increasing relevance. Thus, our approach is not designed to detect cell types in which reduced expression of a specific gene predisposes to obesity. Fourth, while studies have shown that the largest amount of variation is explained by organ rather than species differences (*Brawand et al., 2011*), that the majority of neuronal genes showed similar laminal patterning between human and mouse cortical samples (*Zeng et al., 2012*), and that broadly defined cell types were conserved between mouse and human (*Hodge et al., 2019*), analyses of human tissues may identify additional cell types critical to obesity development. Finally, given the dependence of CELLECT results on other cell types in the given datasets, we, generally, recommend running a 'tiered' prioritization strategy for CELLECT, where one preferably starts with analyzing body-wide or organ-wide transcriptional atlases and then turns to more tissue-centric datasets. While the high polygenicity of obesity and the inaccessibility of the human brain complicate approaches to further establish the enriched cell types' relevance in human obesity, we believe that combinations of functional imaging techniques, postmortem single-nucleus analyses, enhancers to gene maps and fine-mapping of BMI GWAS loci will be crucial to better understand their role in human obesity.

## Strategies for follow-up

Having identified GWAS-enriched cell populations only marks the start of the journey towards understanding how genetic variants render us susceptible to obesity. Two key questions marking the outset of this journey are; What is the subset of associated GWAS variants acting through the enriched cell populations and what are the regulatory elements and effector genes (candidate causal genes) through which these variants exert their effects? And, how is the given cell population affecting physiology and downstream risk of obesity? Given that CELLECT is not specifically designed to identify effector genes but rather intended to identify cell populations enriching for GWAS signal, we suggest to address these questions by focusing on (a) identifying the set of candidate causal variants and effector genes conferring risk through the focal cell population, and (b) directly validating the relevance of the focal cell population under relevant physiological and pharmacological conditions.

Fulco et al. recently proposed an elegant model to map enhancers to effector genes in a given cell type (*Fulco et al., 2019*). Their so-called activity-by-contact model leverages single-cell chromatin accessibility and enhancer activity data to identify cell type-specific enhancers and their target genes. For the focal cell population, such an enhancer-gene map, when integrated with credible sets of fine-mapped GWAS variants, would bring forward a set of testable hypotheses on how a set of candidate causal variants act through a set of specific enhancers and effector genes to impact obesity (or any other disease of interest). Additional confidence could be gained by adding in computational gene prioritization approaches such as DEPICT or MAGMA, for example by up-weighing effector genes that are predicted to be functionally similar to candidate effector genes in the other relevant cell populations. Given species-specific differences in gene regulation, these analyses would need to be performed in animal models with at least partly conserved gene regulatory architectures, human postmortem brain samples (ideally obtained from relevant cases and controls) and/or in induced pluripotent stem cells models (ideally selected from individuals with relevant polygenic backgrounds).

Given the challenges typically encountered in the journey aimed at identifying causal variants and effector genes underlying obesity (for successful examples see *Claussnitzer et al., 2015*; *Smemo et al., 2014*), we suggest in parallel to leverage transgenic animal models to directly assess the relevance of the focal cell in obesity. Given that CELLEX provides marker genes specifically

marking the focal cell population and that all enriched cell populations were of neuronal origin, transgenic animal model techniques such as designer receptors exclusively activated by designer drugs (DREADD)-based chemogenetic tools for activation or inhibition of neurons, transgenic techniques for cell ablation, and fiber photometry techniques for real-time monitoring the impact of relevant physiological environments or pharmacological treatments on the focal cell population, are well-positioned to provide relevant insights into the role of the given cell type in the control of energy homeostasis.

### Relevance to human obesity

Despite these limitations, several lines of evidence suggest that the cell types identified herein to be enriched for BMI GWAS signal are relevant to human obesity. First, weight gain is the most pronounced side effect of subthalamic nucleus deep brain stimulation used to treat Parkinson patients (*Limousin and Foltynie, 2019*), an adverse side effect that may involve the DEINH3 cell type mapping to the subthalamic nucleus. Second, lorcaserin (Belviq), an anti-obesity drug, acts on the 5-HTR2C receptor to enhance serotonin signaling. Third, at the genetic level BMI is significantly correlated with attention deficit/hyperactivity disorder (ADHD) (*Demontis et al., 2019*), and growing evidence points to links between ADHD and eating disorders. For example, lisdexamfetamine (Vyvanse), a medication used to treat ADHD, is also used to treat binge eating (*McElroy et al., 2016*), while the ADHD medication methylphenidate (Ritalin) is known to reduce appetite (*Faraone et al., 2008*). These pharmacological observations suggest that the shared heritability of BMI and ADHD may involve pleiotropic gene variants acting through dorsal midbrain pathways. Fourth, genetic predisposition to obesity is protective to feelings of worry (*Millard et al., 2019*), supporting our findings that these two traits are potentially acting through overlapping cell types in the dorsal midbrain. Finally, BMI variants associated with BMI in a GWAS conducted in Japanese individuals enriched most highly for enhancers active in the hippocampus (*Akiyama et al., 2017*) and maternal obesity is associated with reduced total hippocampal volume in reduced CA3 volume in children (*Page et al., 2018*). Together these observations support a model in which integration of sensory signals, the dopamine system and memory are likely to play key roles in regulating susceptibility to obesity.

In conclusion, our results implicate specific brain nuclei regulating integration of sensory stimuli, learning and memory in human obesity and provide testable hypotheses for mechanistic follow-up studies. Our methodological framework provides a salient example of how human genetics data can be integrated with murine scRNA-data to identify and map components of brain circuits underlying obesity. We provide easy to use computational toolkits, CELLECT and CELLEX, which we envision will greatly facilitate future functional interpretation of genetic association data.

## Materials and methods

### GWAS

For our primary analysis we obtained BMI GWAS summary statistics performed in UK Biobank participants ($N_{max}$ = 457,824) (*Loh et al., 2018*). To examine the robustness of our results to changes in GWAS cohort size, we performed secondary analyses on BMI GWAS summary statistics from two meta-analyses described in *Yengo et al., 2018* $N_{max}$ = 795,640, UK Biobank and GIANT cohorts and *Locke et al., 2015* ($N_{max}$ = 322,154; European subset). We note that these two studies include individuals genotyped on custom array chips (Illumina Metabochip), which violate certain assumptions of S-LDSC, however, we show that this has a negligible effect on our results. *Figure 2—source data 1* provides the full list of GWAS summary statistics analyzed here. We used the script 'munge_sumstats.py' (LDSC v1.0.0, see URLs) to prepare all GWAS summary statistics. All prepared statistics were restricted to HapMap3 single nucleotide polymorphisms (SNPs), excluding SNPs in the major histocompatibility complex region (chr6:25Mb-34Mb).

### Single-cell RNA-seq datasets

For the Tabula Muris dataset (*Tabula Muris Consortium et al., 2018*; SmartSeq2 protocol) cell types were defined as unique combinations of cell ontology and organ annotation (for example, 'Lung-Endothelial_cell') resulting in n = 115 cell type annotations (of which one was defined as neuronal).

For the Mouse Nervous System dataset (*Zeisel et al., 2018*; 10x Genomics protocol), we used the 'ClusterName' option as cell type annotations (n = 265, of which 214 were defined as neuronal). For the hypothalamus, we leveraged datasets from six studies:

- Arc-ME: Arcuate nucleus and median eminence complex (*Campbell et al., 2017* DropSeq protocol). We used the 'Subcluster' annotations (n = 65, of which 34 were defined as neuronal).
- POA: Preoptic area (*Moffitt et al., 2018* 10x Genomics protocol) dataset. We used the 'Non-neuronal.cluster.(determined.from.clustering.of.all.cells)' annotations for non-neuronal cell types (n = 21) and the 'Neuronal.cluster.(determined.from.clustering.of.inhibitory.or.excitatory. neurons)' annotation for neuronal cell types (n = 66).
- LHA: Lateral Hypothalamic Area (*Mickelsen et al., 2019*, 10x Genomics protocol). We used the 'dbCluster' annotations (n = 43, 30 neuronal).
- VMH: Ventromedial Hypothalamus (*Kim et al., 2019a* SMART-seq and 10x Genomics protocols). We used the 'smart_seq_cluster_label' annotations for the SMART-seq dataset (n = 48, of which 40 were defined as neuronal) and the 'tv_cluster_label' annotation for the 10x Genomics dataset (n = 29, all neuronal).
- HYPC: Pan hypothalamus (*Chen et al., 2017*, DropSeq protocol). We used the 'SVM_clusterID' annotations (n = 45, 34 neuronal).
- HYPR: Pan hypothalamus (*Romanov et al., 2017*, Fluidigm C1 protocol) dataset, we used the 'level1 class' annotation for non-neuronal populations (n = 6) and the 'level2 class (neurons only)' annotation for neurons (n = 54).

Code to download and reproduce preprocessing of all datasets are available via GitHub (see URLs). *Figure 2—source data 2*, *Figure 3—source data 1* and *Figure 5—source data 1* list cell type annotations, the number of cells per cell type and relevant metadata for the *Tabula Muris*, *Mouse Nervous System* and hypothalamus datasets (for each hypothalamus dataset we list the cell type labels used in this study as well as the cell type labels used in the original studies).

## Single-cell RNA-seq data pre-processing

For each dataset, we began with a matrix of gene expression values. We normalized expression values to a common transcript count (with $n$ = 10,000 transcripts as a scaling factor) and applied log-transformation ($log(x + 1)$). Next we excluded 'sporadically' expressed genes following the approach described in *Skene et al., 2018* using a one-way ANOVA with cell type annotations as the grouping factor and excluding all genes with p>$10^{-5}$. We mapped mouse genes to orthologous human genes using Ensembl (v. 91), keeping only 1–1 mapping orthologs.

## Cell type labels

For the *Mouse Nervous System* dataset, we used (*Zeisel et al., 2018*) cell type annotations: the first two letters in each cell type abbreviation denote the developmental compartment (ME, mesencephalon; DE, diencephalon; TE, telencephalon), letters three to five denote the neurotransmitter type (INH, inhibitory; GLU, glutamatergic) and the numerical suffix represents an arbitrary number assigned to the given cell type. Likewise, for the Tabular Muris dataset, we used the cell type labels as reported in their paper. For the six hypothalamic datasets, we added a label to allow the reader to more easily understand, from which part of the hypothalamus a given cell type was sampled in the original study ('ARCME', arcuate nucleus median eminence complex; 'HYPC', hypothalamus *Chen et al., 2017*; 'HYPR', hypothalamus *Romanov et al., 2017*; 'LHA', lateral hypothalamus; 'POA', preoptic area; 'VMH', ventromedial nucleus) and the cell type it was annotated to in the original work.

## Mouse nervous system neurotransmitter annotation

We used the 'Neurotransmitter' column of the cell type metadata (from the mousebrain.org website) to group neuronal cell types into six neurotransmitter classes (transmitter listed in parenthesis): 'excitatory' (glutamate), 'inhibitory' (GABA or glycine), 'monoamines' (adrenaline, noradrenaline, dopamine, serotonin), 'acetylcholine' (acetylcholine), 'nitric oxide' (nitric oxide) and 'undefined' for neurons not matching these classes or without neurotransmitter data. When cell types were annotated with multiple transmitter classes in the 'Neurotransmitter' column (e.g. glutamate and adrenaline), excitatory or inhibitory class took precedence in our assignment.

## CELLEX expression specificity

See Appendix 2 for a discussion on ES calculations, assumptions and limitations. CELLEX version 1.0.0 was used to produce all results reported in this manuscript. See URLs for a ready-to-use Python implementation of CELLEX. We calculated expression specificity separately for the Tabula Muris, the Mouse Nervous System and each of the hypothalamus datasets. Cell type expression specificity weights ($ES_w$) were calculated using four ES metrics her referred to us as *Gene Enrichment Score (GES)* (**Zeisel et al., 2018**), *Expression Proportion (EP)* (**Skene et al., 2018**), *Normalized Specificity Index (NSI)* (**Dougherty et al., 2010**) and *Differential Expression T-statistic (DET)*. The mathematical formulas for the ES metrics can be found in Appendix 2. For each ES metric, we separately computed gene-specific $ES_w$ values before averaging them into a single ES estimate ($ES_\mu$) using the following steps:

1. For each cell type we determined the set of specifically expressed genes, $G_s$, by testing the null hypothesis that a gene is no more specific to a given cell type than to cells selected at random. We computed empirical $P$-values of ES weights by comparing observed weights for cell type $c$ to 'null' weights obtained by sampling the dataset's cell type annotations (including annotations from cell type $c$ without replacement).
2. For each cell type we calculated $ES_{w*}$ representing the genes' score of being specifically expressed in a given cell type. We assumed that each cell type has a set of specifically expressed genes exhibiting a linearly increasing score reflecting its expression specificity. We modeled this linearity assumption by rank normalizing $ES_w$ for genes, $g$, in $G_s$:
   $$ES_{w*}(g) = rank_g(ES_w(g))/|G_s| \, if \, g \in G_s$$
   $$ES_{w*}(g) = 0 \, if \, g \notin G_s$$
   Note that $ES_{w*}$ are scaled such that $ES_{w*} \in [0, 1]$.
3. For each cell type, we calculated $ES_\mu$, representing a gene's score of being specifically expressed in a given cell type, by taking the mean $ES_{w*}$ across all ES metrics (we here assume equal weighing of ES metrics).

We use '*ES genes*' to denote the set of genes with $ES_\mu > 0$ for a given cell type. Hence, all genes being part of at least one $G_s$ for a specific cell type will be included in the set of *ES genes* for this cell type. *Figure 3—source data 3* and *Figure 5—source data 3* show the number of *ES genes* for the BMI GWAS-enriched *Mouse Nervous System* and hypothalamus cell types. We note that ES genes include genes that were not only strictly specifically expressed (only expressed in the cell type) but also those that were loosely specifically expressed (i.e. have higher expression in the cell type). All cell type enrichment results were computed based on the $ES_\mu$ estimates. CELLEX can take count data as well as transcripts per million-normalized data as input.

## Expression specificity of known marker genes

First, to validate that our ES approach was able to delineate cell type-specific genes, we, for each of the four ES metrics, computed $ES_w$ estimates across four cell types with genes known to be specifically expressed in these cell types, namely hepatocytes (Apoa2), pancreatic alpha-cells (Gcg), striatum medium spiny neurons (Drd2) and mediobasal hypothalamic agouti related peptide (Agrp)-expressing neurons (Agrp). The four $ES_w$ metrics and the combined $ES_\mu$ metric correctly ranked the relevant genes at the top (*Figure 1d*). Conversely, plotting $ES_\mu$ values for these four genes across all cell types revealed that hepatocytes and alpha-cells exhibited the highest $ES_\mu$ for Apoa2 and Gcg, respectively, and that medium spiny neurons and Agrp-positive neurons exhibited the highest $ES_\mu$ for Drd2 and Agrp, respectively (*Figure 1e*).

## CELLECT genetic prioritization of trait-relevant cell types

See Appendix 1 for adiscussion on assumptions and limitations. CELLECT version 1.0.0 was used to produce all results reported in this manuscript. See URLs for a ready-to-use Python implementation of CELLECT. Throughout this paper, we report CELLECT cell type prioritization results using S-LDSC, as this model has been shown to produce robust results with properly controlled type I error (**Finucane et al., 2018**). Cell type prioritization results using MAGMA (**de Leeuw et al., 2015**) can be found in *Figure 3—figure supplement 3b* and *Figure 3—source data 7*.

## Stratified linkage disequilibrium score regression

We used stratified S-LDSC (v. 1.0.0, URLs) to prioritize cell types after transforming cell type $ES_\mu$ vectors into S-LDSC annotations. Running S-LDSC with custom annotations follows three steps: generation of annotation files, computation of annotation LD scores and fitting of annotation model coefficients. We created annotations for each cell type by assigning genes' $ES_\mu$ values to genetic variants utilizing a 100 kilobase (kb) window of the genes' transcribed regions. Fulco et al. showed that most enhancers are located within 100 kb of their target promoters (*Fulco et al., 2019*). When a variant overlapped with multiple genes within the 100 kb window, we assigned the maximum $ES_\mu$ value. The relatively large window size was chosen to capture effects of nearby regulatory variants, as the majority of trait-associated variants have been shown to be located in non-coding regions (*Gusev et al., 2014*). Our results were robust to changes in window size (data not shown), consistent with previous work (*Skene et al., 2018*; *Finucane et al., 2018*; *Kim et al., 2019b*). Following the recommendation in *Finucane et al., 2018*, we constructed an 'all genes' annotation for each expression dataset, by assigning the value 1 to variants within 100 kb windows of all genes in the dataset. We used hg19 (Ensembl v. 91) as the reference genome for genetic variant and gene chromosomal positions. When constructing annotations, we used same 1000 Genomes Project SNPs (*Abecasis et al., 2012*) as in the default baseline model used in S-LDSC. Next, we computed LD Scores for HapMap3 SNPs (*Altshuler et al., 2010*) for each annotation using the recommended settings.

For the primary cell type prioritization analysis, we jointly fit the following annotations: (i) the cell type annotation; (ii) all genes annotation (iii) the baseline model (v1.1). For cell type conditional analysis (*Figure 4—figure supplement 1*) we added (iv) the cell type annotation conditioned on when fitting the model.

We ran S-LDSC with default settings and the workflow recommended by the authors. We reported p-values for the one-tailed test of positive association between for trait heritability and cell type annotation $ES_\mu$. We note that the correlation structure among $ES_\mu$ for cell type annotations can lead to a distribution of p-values that is highly non-uniform (*Finucane et al., 2018*). Highly significant p-values occur due to correlated cell types with true signal, whereas cell types negatively correlated with the true signal have p-values near 1. For all results, we used Bonferroni correction within a trait and dataset to control the FWER. We report the regression effect size estimate for each cell type (source data: 'Coefficient' column), which represents the change in per-SNP heritability due to the given cell type annotation, beyond what is explained by the set of all genes and baseline model. We also report standard errors of effect sizes ('Coefficient std error' column), computed using a block jackknife (*Finucane et al., 2015*). Finally, we report the 'annotation size' for each cell type, that measures the proportion of SNPs covered by the cell type annotation (0 means no SNPs were covered by the annotation; 1 means all SNPs were covered). Annotation size was computed as the mean of the cell type annotation.

## S-LDSC heritability analysis

All S-LDSC heritability analyses and reported effect size estimates were obtained on the observed heritability scale, with the exception of heritability estimates for case-control traits shown in the barplots of *Figure 2a* and *Figure 3b*. Here, we report heritability estimates on the liability scale using population prevalences listed in *Figure 2—source data 1*. (The liability scale is needed when the aim of heritability analysis is to compare heritability estimates across traits. On a liability scale the case-control trait is treated as if it has an underlying continuous liability, and then the heritability of that continuous liability is quantified.) To interpret the heritability explained by our continuous-valued $ES_\mu$ cell type annotations, we estimated the heritability of each $ES_\mu$ quintile. We modified the script 'quantile_M.pl' (from the LDSC package) to compute heritability enrichment for five equally spaced intervals of the cell types $ES_\mu$ annotations: (0–0.2), (0.2–0.4), (0.4–0.6), (0.6–0.8), (0.8–1), as well as the interval including zero values only ([0–0]).

## MAGMA cell type prioritization

To assess the robustness of the SNP-level S-LDSC cell type prioritization, we used an alternative gene-level approach inspired by *Skene et al., 2018* and tested the association of gene-level BMI association statistics with cell type $ES_\mu$ using MAGMA (v1.07a) (*de Leeuw et al., 2015*). MAGMA

was run with default settings to obtain gene-level association statistics calculated by combining SNP association p-values within genes and their flanking 100 kb windows into gene-level Z-statistics, while accounting for LD (computed using the 1000 Genomes Project phase 3 European panel; *Abecasis et al., 2012*). Gene-level Z-statistic were corrected for the default MAGMA covariates: gene size, gene density (a measure of within-gene LD) and inverse mean minor allele count, as well the log value of these variables. Next, we used the R statistical language to fit a linear regression model using MAGMA gene-level Z-statistics as the dependent variable and cell type $ES_\mu$ as the independent variable. We report cell type prioritization p-values (from the linear regression model) as the positive contribution of cell type $ES_\mu$ regression coefficient to BMI gene-level Z-statistics (one-sided test).

## Cell type geneset enrichment analaysis

To assess cell type enrichment of genesets associated with obesity, we tested if members of the obesity geneset exhibited higher expression specificity ($ES_\mu$) in a given cell type than non-members of the geneset (all other genes in the dataset). Specifically, we used a Wilcoxon rank sum test with continuity correction to obtain one-sided geneset enrichment p-values. We controlled the FWER using the Bonferroni method calculated over all cell types and the rare variant obesity geneset tested. As a precaution against unknown confounders, we also computed empirical p-values by permuting the expression specificity gene labels 10,000 times to obtain 'null genesets' of identical size, and obtained near-identical results (data not shown). We obtained genes with rare coding variants associated with obesity (n = 13 genes) and genes implicated in early onset- and extreme obesity from Turcot et al. Table 1 and Supplementary Table 21, respectively. We combined these genes into a single set of 23 high-confidence obesity genes.

## Cell type gene co-expression networks

We identified cell type gene co-expression networks using robust weighted gene correlation network analysis (rWGCNA) framework proposed by *Langfelder and Horvath, 2008*. To identify gene co-expression networks (or *gene modules*) operating within a cell type, the input to WGCNA is expression data for individual cell types. Briefly our framework consisted of the following steps:

1. We normalized the raw expression values to a common transcript count (with n = 10,000 transcripts as a scaling factor), log-transformed the normalized counts (log(x+1)), and centered and scaled each gene's expression to Z-scores. Cell clusters with fewer than 50 cells were omitted, and genes expressed in fewer than 20 cells were removed. We then used PCA to select the top 5000 highly loading genes on the first 120 principal components. We mapped mouse genes to orthologous human genes using Ensembl (v. 91), keeping only 1–1 mapping orthologs.
2. We then used hierarchical clustering and hybrid tree cutting algorithms to identify gene modules. Module eigengenes, which summarize module expression in a single vector, were computed and used to identify and merge highly correlated modules.
3. Finally, we computed gene-module correlations (kMEs), a measure of gene-module membership, filtering out any genes which were not significantly associated with their allocated module after correcting for multiple testing using the Benjamini-Hochberg method.

## Genetic prioritization of cell type co-expression networks

Genetic prioritization of WGCNA gene modules followed the same framework as for prioritizing cell types. That is, we used S-LDSC controlling for the baseline and 'all genes' annotations. Gene modules annotations were constructed by assigning the module genes' kME values to variants within a 100 kb window of the genes' transcribed regions. We restricted modules to contain at least 10 genes and at most 500 genes (removing 8 out of 571 modules), because S-LDSC is not well-equipped for prioritizing annotations that span very small proportion of the genome, and unspecific modules with a large number of weakly connected genes may have limited biological relevance.

## Co-expression network visualizations

To create the network visualization of the cell type rWGCNA gene modules (*Figure 3—figure supplement 2b*), we computed the Pearson's correlation between module kME values (a measure of

gene-module membership) and generate a weighted graph between modules using the positive correlation coefficients only. To create the network visualization of the M1 gene module (*Figure 3—figure supplement 2c*), we computed the Pearson's correlation between genes within the module, using expression data from the cell type in which the module was identified (MEINH2). We then generate a weighted graph between genes using the positive correlation coefficients only. We then mapped MAGMA BMI gene-level Z-statistics (calculated using 100 kb windows, as described above) onto the network as node sizes. All networks were visualized using the R package 'ggraph' with weighted Fruchterman-Reingold force-directed layout.

## Cell type enrichment of co-expressed gene networks

To assess if gene modules were enriched in the expression specific genes of specific cell types, we tested if module gene members exhibited higher expression specificity ($ES_\mu$) in the given cell type than non-members of the module (all other genes in the dataset). We obtained one-sided enrichment p-values using the Mann-Whitney U test. We controlled the FDR by using the Bonferroni method calculated over gene modules tested.

## Tests for confounding factors and null GWAS construction

In order to test for technical bias in CELLECT genetic enrichment scores, we prioritized cell types using GWAS based on randomly distributed phenotypes ('null GWAS'). We computed 1000 GWAS based on 1000 Genomes Project Phase three genotyping data and simulated Gaussian phenotypes randomly drawn from a N(0,1) distribution with no genetic bias. We then performed genetic prioritization across 115 cell types in the Tabula Muris dataset using CELLECT with S-LDSC for each null GWAS.

S-LDSC prioritization p-values, which for null GWAS tend toward a uniform distribution, showed a slight enrichment for *P*-values closer to 1, and a slight depletion close to 0. To verify that CELLECT genetic prioritization p-values were not correlated with technical factors, we computed the Pearson correlation between the $-\log_{10}$(S-LDSC p-value) for a cell type and the number of cells, median number of genes expressed, and median number of UMIs, respectively for each null GWAS. We used a two-sided t-test to identify significant deviations from the expected mean correlation of zero.

## The genotype-tissue expression consortium data and analysis

The genotype-tissue expression version eight gene expression read counts were obtained from their portal (download date 6 May 2020). An initial set of 17,382 RNA-seq samples were filtered on quality indicators using the same cutoffs as in *GTEx Consortium et al., 2017*. Next, to identify and remove outliers, we used an approach similar to that of *Wright et al., 2014*: within each tissue-type (SMTSD annotation), we computed the mean Pearson correlations of each sample to the others. We then removed any samples whose expression profile had a mean correlation falling below the first quartile by more than 1.5 times the interquartile range within that tissue-type, leaving 16,027 samples from 946 donors. Genes were then filtered, again using the cutoff from *GTEx Consortium et al., 2017*, that is keeping genes with at least six reads in at least 10 samples. To ensure positive expression values as required by CELLEX, and given that common batch-correction techniques typically incur partly negative expression values, we did not perform batch correction. The filtered gene read counts were normalized within each broad tissue-type (SMTS annotation) using the *DESeqDataSetFromMatrix()*, *estimateSizeFactors()* and *counts()* commands from the DESeq2 R package (v1.22.2) (*Love et al., 2014*). Finally, normalized counts were log-transformed ($\log_2(x+1)$), gene version number suffixes were removed from the GENCODE gene names, and samples were grouped by SMTSD annotations for downstream analysis with CELLEX and CELLECT.

## Code availability

CELLECT toolkit is available at *Timshel, 2020*; https://github.com/perslab/CELLECT (copy archived at https://github.com/elifesciences-publications/timshel-2020). CELLEX is available at https://github.com/perslab/CELLEX. Open source software implementations of CELLECT and CELLEX will be made available upon publication. Code to reproduce analyses, figures and tables for this manuscript is available at https://github.com/perslab/timshel-2020.

## URLs

- LDSC: https://github.com/bulik/ldsc
- MAGMA: https://ctg.cncr.nl/software/magma
- Robust WGCNA pipeline: https://github.com/perslab/wgcna-toolbox
- Genotype-Tissue Expression Consortium portal: https://gtexportal.org/home/datasets

## Acknowledgements

Novo Nordisk Foundation Center for Basic Metabolic Research is an independent Research Center, based at the University of Copenhagen, Denmark and partially funded by an unconditional donation from the Novo Nordisk Foundation (www.cbmr.ku.dk) (Grant number NNF18CC0034900). THP acknowledges the Novo Nordisk Foundation (Grant number NNF16OC0021496) and the Lundbeck Foundation (Grant number R19020143904). PNT acknowledges the Danish Ministry of Higher Education and Science for the Elite Research PhD scholarship.

We gratefully acknowledge Diego Calderon for helpful discussions on genetic prioritization models; Steven Gazal for support with LDSC heritability enrichment; Christiaan de Leeuw for support on MAGMA; Stephen Quake, Spyros Darmanis and the Biohub team for providing pre-publication access to the Tabula Muris dataset; Michael W Schwartz, Thorkild IA Sørensen, Lars Ängquist and Dylan M Rausch for helpful inputs on neuroendocrinology and obesity; Tobias Stannius, Ben Nielsen, Tobi Alegbe, Petar V Todorov and Liubov Pashkova for improving the CELLECT and CELLEX software.

## Additional information

### Funding

| Funder | Grant reference number | Author |
| --- | --- | --- |
| Novo Nordisk Foundation | NNF16OC0021496 | Tune H Pers |
| Lundbeck Foundation | R19020143904 | Tune H Pers |

The funders had no role in study design, data collection and interpretation, or the decision to submit the work for publication.

### Author contributions

Pascal N Timshel, Conceptualization, Resources, Data curation, Software, Formal analysis, Supervision, Validation, Investigation, Visualization, Methodology, Writing - original draft, Writing - review and editing; Jonatan J Thompson, Resources, Data curation, Software, Formal analysis, Visualization, Methodology, Writing - review and editing; Tune H Pers, Conceptualization, Resources, Data curation, Supervision, Funding acquisition, Validation, Investigation, Visualization, Methodology, Writing - original draft, Project administration, Writing - review and editing

### Author ORCIDs

Pascal N Timshel (iD) https://orcid.org/0000-0002-0352-4323
Jonatan J Thompson (iD) http://orcid.org/0000-0003-4969-9927
Tune H Pers (iD) https://orcid.org/0000-0003-0207-4831

### Decision letter and Author response

Decision letter https://doi.org/10.7554/eLife.55851.sa1
Author response https://doi.org/10.7554/eLife.55851.sa2

## Additional files

### Supplementary files

- Transparent reporting form

## Data availability

All data generated or analysed during this study are included in the manuscript, supporting files and on https://github.com/perslab/timshel-2020 (copy archived at https://github.com/elifesciences-publications/timshel-2020).

The following previously published datasets were used:

| Author(s) | Year | Dataset title | Dataset URL | Database and Identifier |
|---|---|---|---|---|
| Gloudemans M, Balliu B | 2018 | GWAS studies | https://github.com/mike-gloudemans/gwas-download | GitHub, gwas-download |
| Romanov RA, Zeisel A, Bakker J, Girach F, Hellysaz A, Tomer R, Alpár A, Mulder J, Clotman F, Keimpema E, Hsueh B, Crow AK, Martens H, Schwindling C, Calvigioni D, Bains JS, Máté Z, Szabó G, Yanagawa Y, Zhang MD, Rendeiro A, Farlik M, Uhlén M, Wulff P, Bock C, Broberger C, Deisseroth K, Hökfelt T, Linnarsson S, Horvath TL, Harkany T | 2017 | Hypothalamus - HYPR | https://www.ncbi.nlm.nih.gov/geo/query/acc.cgi?acc=GSE74672 | NCBI Gene Expression Omnibus, GSE74672 |
| Kim D-W, Yao Z, Graybuck LT, Kim TK, Nguyen TN, Smith KA, Fong O, Yi L, Koulena N, Pierson N, Shah S, Lo L, Pool A-H, Oka Y, Pachter L, Cai L, Tasic B, Zeng H, Anderson DJ | 2019 | Hypothalamus - VMH | https://doi.org/10.17632/ypx3sw2f7c.1 | Mendeley Data, 10.17632/ypx3sw2f7c.1 |
| Chen R, Wu X, Jiang L, Zhang Y | 2017 | Hypothalamus - HYPC | https://www.ncbi.nlm.nih.gov/geo/query/acc.cgi?acc=GSE87544 | NCBI Gene Expression Omnibus, GSE87544 |
| Moffitt JR, Bambah-Mukku D, Eichhorn SW, Vaughn E, Shekhar K, Perez JD, Rubinstein ND, Hao J, Regev A, Dulac C, Zhuang X | 2018 | Hypothalamus - POA | https://www.ncbi.nlm.nih.gov/geo/query/acc.cgi?acc=GSE113576 | NCBI Gene Expression Omnibus, GSE113576 |
| Campbell JN, Macosko EZ, Fenselau H, Pers TH, Lyubetskaya A, Tenen D, Goldman M, Verstegen AMJ, Resch JM, McCarroll SA, Rosen ED, Lowell BB, Tsai LT | 2017 | Hypothalamus - ARCME | https://www.ncbi.nlm.nih.gov/geo/query/acc.cgi?acc=GSE93374 | NCBI Gene Expression Omnibus, GSE93374 |
| Mickelsen LE, Bolisetty M, Chimileski BR, Fujita A, Beltrami EJ, Costanzo JT, Naparstek JR, Robson P, Jackson AC | 2019 | Hypothalamus - LHA | https://www.ncbi.nlm.nih.gov/geo/query/acc.cgi?acc=GSE125065 | NCBI Gene Expression Omnibus, GSE125065 |

| The Tabula Muris Consortium | 2018 | Tabula Muris | https://www.ncbi.nlm. nih.gov/geo/query/acc. cgi?acc=GSE109774 | NCBI Gene Expression Omnibus, GSE109774 |
| --- | --- | --- | --- | --- |
| Zeisel A, Hochgerner H, Lönnerberg P, Johnsson A, Memic F, Zwan J, Häring M, Braun E, Borm LE, Manno GL, Codeluppi S, Furlan A, Lee K, Skene N, Harris KD, Hjerling-Leffler J, Arenas E, Ernfors P, Linnarsson S | 2018 | Mouse Nervous System | https://www.ncbi.nlm. nih.gov/sra/SRP135960 | NCBI Sequence Read Archive, SRP135960 |

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

## Appendix 1

## CELLECT cell type prioritization

Introduction

Here we discuss the detailed methods and limitations of the CELLECT framework.

### The relevance of using mouse scRNA-seq datasets

Our BMI cell type prioritization analysis was performed using mouse scRNA-seq atlases. We here discuss the relevance of using mouse scRNA-seq datasets to define cell types for genetic prioritization for complex human traits.

Previous studies have compared the conservation of tissue gene expression. *Brawand et al., 2011* used RNA-seq expression data across multiple organs (cortex, cerebellum, heart, kidney, liver, and testis) and 10 mammalian species (incl. human) and found the largest amount of variation was explained by organ rather than species differences.

Previous studies have assessed the convergence of mouse and human central nervous system (CNS) gene expression using gene co-expression analysis (*Hawrylycz et al., 2015*; *Kelley et al., 2018*; *Miller et al., 2010*) and found weaker conservation of glial co-expression modules than neuronal co-expression modules. In situ hybridization studies have reported that the majority of genes (79%) showed similar cortical laminar patterning (*Zeng et al., 2012*). Along those lines, recent scRNA-seq data from mouse and human midbrain found that cell types and gene expression levels were generally conserved across species (*La Manno et al., 2016*).

The current most extensive study of CNS cell type conservation compared single-nucleus expression data from human and mouse cerebral cortex (*Hodge et al., 2019*) and found that broadly defined cell types were conserved between mouse and human. However, they identified important differences between cell type proportions and expression of specific genes, including cell type marker genes exhibiting up to 10-fold expression differences.

In conclusion, although critical differences between mouse and human CNS gene expression data have been identified, the broad expression patterns are likely to be conserved. Moreover, glial cell types are more likely to exhibit weaker conservation compared to neuronal cell types. We believe our genetic prioritization of likely etiologic cell types is more likely to suffer from false negatives (cell types not prioritized because of lack of relevant human expression data) rather than false positives (spuriously enriched cell types among our positive results).

### Choice of window size and position for connecting SNPs and genes

An important step in the CELLECT pipeline is assigning gene $ES_\mu$ values to SNPs. As the majority of trait-associated SNPs are located in non-coding regions (*Gusev et al., 2014*), it is desirable to select a window size that maps the majority of regulatory GWAS variants to their proximal genes. Although our results were largely robust to changes in window size and consistent with previous work (*Finucane et al., 2018*; *Kim et al., 2019a*; *Skene et al., 2018*), we note that the SNP-to-gene mapping remains a critically important step that should be updated in subsequent versions of CELLECT.

In this work, we used the same window size as used in *Finucane et al., 2018*, that is, assigning genes' $ES_\mu$ values to SNPs within a 100 kb window on either side of a gene's transcribed regions. A recent large eQTL analysis in blood from >31,000 individuals found that 92% of the lead cis-expression quantitative trait loci (eQTL) SNPs mapped within 100 kb of the gene (*Võsa et al., 2018*), suggesting that our mapping is likely to capture the majority of cis-regulatory variants. Consistent with this, *Gasperini et al., 2018* used CRISPR/Cas9 followed by scRNA-seq to identify CRISPR/Cas9-induced eQTLs from >47,000 human cell line cells and found that regulatory variants were separated from the TSS of their target genes by a median distance of 34.3 kb. Finally, work by Fulco et al. reports that most enhancers are located within 100 kb of the target promoters (*Fulco et al., 2019*).

## Limitations of CELLECT

### Linear relationship between expression specificity and trait heritability

The overall assumption behind our approach is that in order for a disease to manifest in a given cell type the set of disease causal genes must be active and expressed in the given cell type. In other words, we assume that high/increased expression and not decreased/lack-of expression of a gene results in disease. This is a strong assumption to make about complex traits and it does not hold for all diseases (e.g. cancer).

Our model assumes a linear effect of cell type expression specificity and trait heritability. Although this assumption may not always hold, it appears to be reasonable in the continuous annotations that we analyzed (*Appendix 4—figure 1*).

We leave it for future work to explore non-linear relationships between expression specificity and trait heritability, and to investigate the effects of specificity for decreased or lack-of gene expression.

### Genetic architecture

The approach assumes that a cell type is etiologic for a particular disease if and only if genetic variants near genes with high expression specificity in the cell type are enriched for heritability. Moreover, the CELLECT cell type prioritization assumes a polygenic trait architecture. Consequently, our approach is unlikely to yield relevant results for traits driven by rare genetic mutations (not covered by GWAS) or traits where the heritability is not mediated by transcriptional differences (i.e. changes related to other molecular modalities such as proteins, posttranslational modifications or the microbiome).

### Common variation

We restricted our analysis to common variants (HapMap3 SNPs,>5% MAF), as S-LDSC has several limitations when applied to rare variants. Prioritizing cell types using a model that includes both common and rare variants could produce different results. We argue that our results are likely to be robust to changes in the allele frequency spectrum. Firstly, we found that cell types enriched for rare variant obesity genes overlapped with the S-LDSC BMI prioritized cell types (based on common variants). Secondly, a recent study by *Zhu and Stephens, 2018* compared the ability of genetic enrichment methods (incl. S-LDSC) to detect the true enrichment signal based on 1000 Genomes Project SNPs and HapMap3 SNPs, and found that all methods (S-LDSC included) produced similar results using the two sets of SNPs as input. Thirdly, rare variants are unlikely to explain the majority of BMI heritability: Gazal et al. estimated the low-frequency variants (MAF <5%) to explain 15% of BMI heritability (*Gazal et al., 2018*) and recent work (*under review*) has reported that variants with MAF <10% might explain as much as 51% of BMI heritability [*Wainschtein et al., 2019*]. In conclusion, cell type prioritization results restricted to common variants are likely to converge with results including rare variants.

### Expression heritability mediated by cis- vs trans-eQTLs

As our model assumes that SNPs *near* genes with high expression specificity in etiologic cell types are enriched for heritability, our model relies on the majority of gene regulatory variants (eQTLs) are located nearby (*cis-acting*) instead of distant (*trans-acting*) to the target gene. That is, we assume that heritability of gene expression can be sufficiently explained by *cis*-acting variation. There are notable examples where the causal regulatory variant act in trans, for example the causal variant located in the first intron of the FTO locus are located >1 Mb from its target regulatory genes IRX3/IRX5 (*Claussnitzer et al., 2015*; *Smemo et al., 2014*), but the question how prevailing *trans*-acting variation is remains unresolved. In support of the sufficiency of *cis*-variation, one study found that *cis*-eQTLs explain a substantial proportion of trait heritability (40–80%) (*Gamazon et al., 2018*). In addition, transcriptome-wide association studies (TWAS) leverage *cis*-eQTLs to predict expression levels with a 60–80% prediction accuracy (*Gamazon et al., 2015*; *Gusev et al., 2016*). In contrast, *Liu et al., 2019* report that up to 60–90% of genetic variance in expression is due to trans-acting

variation. We acknowledge that trans-acting effects are likely to play an important role in gene expression heritability, but despite promising efforts (*Fulco et al., 2019*) cell type-specific enhancer to gene maps have not been constructed yet and hence we based CELLECT on cis-regulatory variants only.

## Future directions

We envision several improvements of our approach. SNP-to-gene mapping could be improved by levering for instance the ABC model propsed by *Fulco et al., 2019* cell types to predict enhancer to promoter maps to assign regulatory variants to genes. Alternatively, SNP-to-gene mapping could be improved by using LD-informed loci definitions centered on SNPs. That is, each SNP would be assigned $ES_\mu$ value based on the genes within the LD defined loci boundaries of the SNP (e.g. genes within the region spanned by $r^2 < 0.7$).

It would be of interest to explore non-linear relationships between expression specificity and heritability and to test whether down-regulated genes contribute to cell type heritability. Such analysis would be possible leverage an extension of LDSC referred to as signed linkage disequilibrium profile regression (*Reshef et al., 2018*), which allows detection of directional effects of signed functional annotations.

Finally, we envision a data-driven approach to select the parameters of our approach, for example SNP-to-gene parameters or non-linear transformation of expression specificity. A genetic trait with known etiologic cell types could be used to select the set of parameters resulting in the most significant prioritization of the known etiologic cell types.

## Levering the omnigenic model to detect disease causal cell types

In the following we attempted to unify our cell type prioritization model with the so-called omnigenic hypothesis proposed by Prichard and colleagues (*Boyle et al., 2017*; *Liu et al., 2019*). We here describe the key assumptions and approach behind CELLECT.

## Key assumptions and observations

Our key assumption is that in order for a disease to manifest in a given tissue or cell type the set of disease causal genes must be active and expressed in the given tissue or cell type. That is, we assume that high/increased expression and not decreased/lack-of expression of a gene results in the given trait or disease (henceforth simply referred to as *disease*). This is a strong assumption to make about complex traits and it may not hold for all diseases (e.g. cancer).

We assume that for a cell type to be causal to a given disease, it should express one or more core genes. We note an important distinction between this assumption and the stronger more commonly used assumption that causal, disease cell types enrich for expression of all core genes (e.g. testing for top expressed cell type genes for enrichment of genes harboring rare variants *Skene et al., 2018*). Because core genes can function in orthologous pathways, we only assume expression of one or more core genes.

We assume that core genes have cell type specific etiologic roles for common complex traits. This assumption is justified by the strong negative selection of mutations in genes with a ubiquitous function broadly affecting cellular function. (Detrimental mutations in genes with non-redundant basic cellular function will not manifest in the population as a common disease.) We note that this only holds true for *heritable common* diseases. for example core driver cancer genes may have basic biological functions as observed with de novo mutations in *TP53*.

We reason that the majority of genes localizing in GWAS loci are peripheral genes that are more likely than other genes to exhibit cell type-specific expression. This assumption is justified by two steps of reasoning. Firstly, peripheral genes can only exert their effect on core genes if they are co-expressed in a cell. They must operate within the same network of expressed genes in a cell (see *Boyle et al., 2017 Figure 4b*). Secondly, GWAS is more well-powered to detect peripheral genes in close proximity to core genes, if a shorter degree of separation between peripheral and core genes increases the effect of the peripheral gene on the core gene (*ibid.*; *Figure 4a*). We note that under the 'small world' network property of gene regulatory networks, most expressed genes in a cell type

are only a few steps from the nearest core gene, possibly making the set of 'peripheral genes in close proximity to core genes' quite large.

## Biological examples of potential mechanisms

It has been shown that impaired signaling from the primary cilia of *MC4R-positive* neurons can cause obesity in humans. In vitro and in vivo work from *Siljee et al., 2018* demonstrated that *MC4R* obesity-causing mutations impair the localization of *MC4R* to the cilia. The discovery of obesity-causal mutations in *ADCY3* provided evidence that *ADCY3* plays a role in human obesity (*Grarup et al., 2018*; *Saeed et al., 2018*). This example may demonstrate how genetic variation in one, herein assumened, peripheral gene, *ADCY3*, can regulate/impair the function of a core gene, in this case, *MC4R*, within the energy-regulating melanocortin signaling pathway. The omnigenic model suggests that there are many yet unknown genetic regulators affecting *MC4R* (potentially also through primary cilia signaling or targeting) and hence contributing to the obesity heritability.

Our results for prioritized cell types could support this distinction between core and peripheral gene. We find that the core gene (*MC4R*) is co-expressed with the peripheral gene (*ADCY3*) (**Appendix 1—figure 1**). *MC4R* is highly specifically expressed and *ADCY3* is moderately specifically expressed.

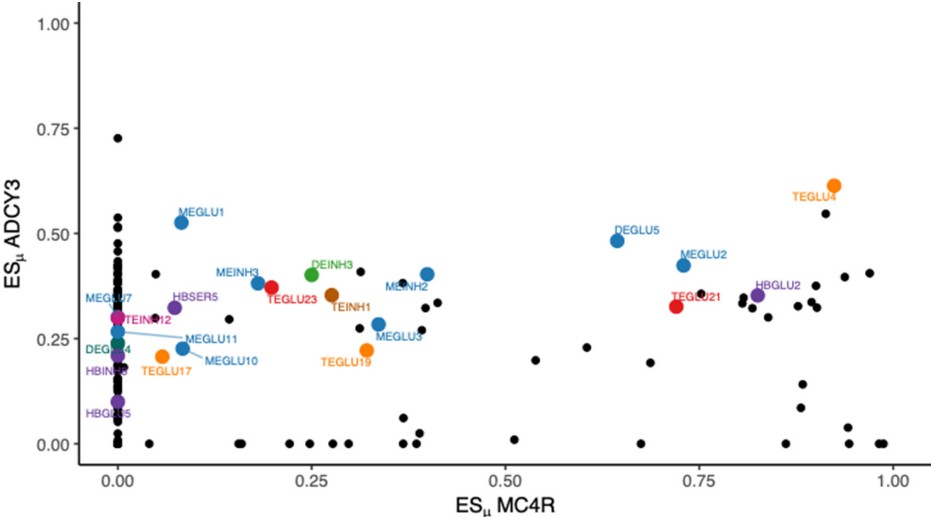

**Appendix 1—figure 1.** Co-specific expression of Mc4r and Adcy3. The co-specific expression of Mc4r and Adcy3 may serve as an example on how certain cell types may co-express a core gene (Mc4r in this example) and peripheral genes (Adcy3 in this example). BMI-prioritized cell types are highlighted in color.

## The model

Our framework, CELLECT uses *continuous* LDSC annotations to identify likely disease causal cell types. Here we explain why using *continuous* (i.e. weighted) annotations is an important improvement over existing studies.

We assume that if a gene is specifically expressed in a given cell type, it is functionally important for that cell type. That is, specifically expressed genes constitute the functionally distinct part of the cell type. For instance, we have shown that *DRD2* is specifically expressed in certain cell types from the midbrain, suggesting the functional role of these cell types in the dopamine reward system. *ES genes* will, by definition, not contain ubiquitous/equally expressed genes (e.g. basic cellular/biological processes). Please also refer to Appendix 2.

Following our above assumptions, for a disease causal cell type we assume the following relationship of cell type specific expression:

$$ES(\text{trait relevant core genes}) \geq ES(\text{trait peripheral genes}) > ES(\text{non trait relevant genes})$$

We are now able to express our model for genetic identification of causal disease cell types.

Formally we model a linear relationship between a gene's disease heritability and cell type expression specificity:

$$Heritability(gene) \sim ES(gene)$$

*Appendix 1—figure 2* shows the concept of how the two above equations can be used to identify likely disease causal cell types.

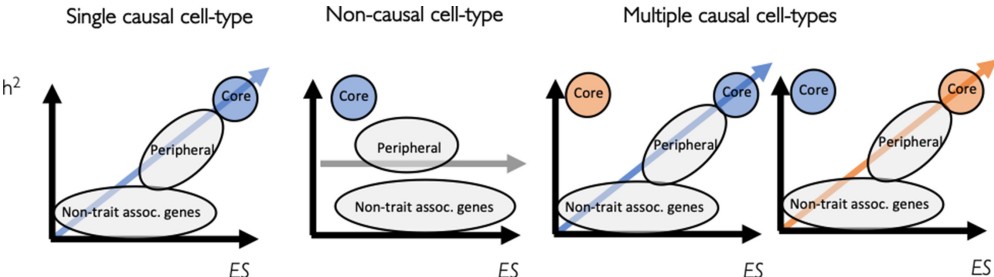

**Appendix 1—figure 2.** Levering the omnigenic model to identify causal cell types using expression specificity. A linear model can be used to identify causal cell types by testing for association between gene expression specificity and gene heritability. Three scenarios are shown: a causal cell type (left), a non-causal cell type (middle) and multiple causal cell types (right).

In support of this model, we showed that top expression specific genes in BMI-prioritized cell types exhibit higher BMI heritability enrichment than non-expression specific genes.

## Summary

We here provide a brief summary of the points discussed in the above sections.

## Model assumptions and observations

- The 'omnigenic' genetic architecture of complex traits states that so-called peripheral genes (peripheral referring to the core molecular function encoded by the *core* genes in the cell type) explain the majority of heritability, and as a consequence, identifying heritability enrichment using only core genes, will fail.
- To identify disease causal cell types, we estimate the disease heritability explained by specifically expressed genes.
- Likely disease causal cell types have one or more core genes specifically expressed *and* the specifically expressed genes enrich for disease heritability.
- Core genes have cell type specific roles and are specifically expressed.
- Peripheral genes are co-expressed with at least one core gene in the disease-causal cell type.
- Peripheral genes are more likely than other genes to be specifically expressed.
- Many peripheral genes are shared among related traits. This leads to a partial overlap between prioritized cell types for related traits. Core genes have little overlap between related traits.

## Results

- We show that the far majority of our prioritized cell types express one or more 'core' obesity genes (as defined by the high-confidence obesity genes geneset). See *Figure 5—figure supplement 3*.
- We show that top expression specific genes in prioritized cell types have higher BMI heritability enrichment than non-specifically expressed genes (*Appendix 4—figure 1*).

## Appendix 2

## CELLEX expression specificity

### Introduction

At the core of using expression data for genetic identification of cell types underlying disease, lies the problem of finding a meaningful vector representation of cell type expression profiles. In our approach, we represent cell types by their expression specificity (ES) profile: a measure of relative gene expression levels. To robustly estimate ES, we developed CELLEX.

Here we provide additional details and limitations of the CELLEX expression specificity framework. We provide a comprehensive benchmark of ES metrics on single-cell RNA-sequencing (scRNA-seq) and show that our combined metric, $ES_\mu$, is most robust than single expression specificity measures. For an ES metric benchmark on bulk data we refer to *Kryuchkova-Mostacci and Robinson-Rechavi, 2016*.

## CELLEX expression specificity

### Notation

We use the term 'ES metric' to describe the given metric used to compute $ES_w$ (see *Appendix 2—figure 1—source data 1* for an overview of the ES metrics used). $ES_w$ are gene-level statistics computed for a given ES metric. $ES_{w*}$ are the genes' likelihood of being specifically expressed given the ES metric. $ES_\mu$ are the genes' marginal likelihood of being specifically expressed.

Note that ES values are computed separately for each dataset. *Appendix 2—figure 1* provides an overview of the steps.

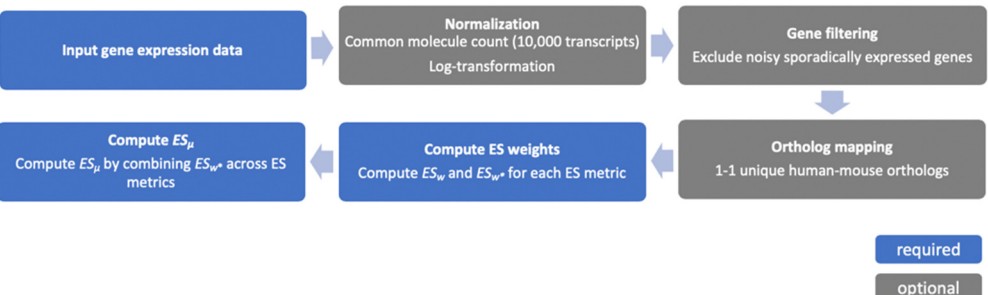

**Appendix 2—figure 1.** Overview of CELLEX expression specificity estimation In this work, we used the optional steps for normalization and gene filtering. We also used ortholog mapping when analyzing mouse scRNA-seq data.

The online version of this article includes the following source data is available for figure 1:

**Appendix 2—figure 1—source data 1.** ES metrics used in CELLEX.

### Expression data pre-processing and normalization

For this work we normalized gene expression values using the default normalization approach for CELLEX (described below). We note that the default CELLEX normalization can be disabled to allow a for data input that has been normalized using a customized approach. The customized normalization approach must not return negative values, as the ES metric calculation assumes that all normalized input values are positive or zero.

### Default normalization

As described above, we scale expression values to a common transcript count with n = 10,000 transcripts as a scaling factor. Next, we apply log-transformation ($x_{log} = log(x + 1)$). We apply this normalization procedure for the following three reasons:

1. Common transcript count makes expression values comparable among cells.
2. Log-normalization is a variance-stabilizing transformation that dampens the effect of the dynamic range. It transforms the expression data into a more well-behaved distribution that is

better approximated by a normal distribution (we later compute ANOVA and t-statistics that assume normality of the data).

3. This normalization procedure is a common, robust and proven-to-work method for scRNA-seq (it is the default normalization technique for standard single-cell analysis packages such as Seurat; *Satija et al., 2015*). This normalization method was also used by the authors of the two primary data sets analyzed in our study: the Mouse Nervous System (*Zeisel et al., 2018*) and Tabula Muris (*Tabula Muris Consortium et al., 2018*) datasets.

Despite the above strengths, we note two limitations of this normalization approach:

- When estimating the common transcript count 'scaling factors' for each cell, we assume each cell has the same number of total molecules. This is an overly simplistic assumption as cell size is *Townes et al., 2019* generally correlated with the amount of mRNA in the cell.
- Log-transformation is an empirical choice for variance-stabilizing transformation. Recently developed generalized linear models might provide better normalization (*Hafemeister and Satija, 2019*; *Townes et al., 2019*).

We leave it for future work to explore additional normalization procedures and approaches to correct for confounding factors between cell types prior to estimating ES weights.

## Gene filtering

We observed that the ES metrics were prone to falsely estimating genes with 'sporadic' gene expression levels as highly expression specific. We excluded these genes to reduce bias in the null expectation for $ES_w$ and hence reduce false positive ES genes. Most notably, the EP metric would estimate genes with very low expression appearing in few number of cells as highly expression specific. To solve this problem, we sought to estimate the background noise level for each gene, enabling us to distinguish between genes with undetectable sporadic expression levels and genes with confident expression levels. Following the approach described in *Skene et al., 2018*, we reasoned that genes with sporadic expression would fail to be statistically differentially expressed in at least one cell type. We modelled this using one-way ANOVA with cell type annotations as the grouping factor and excluded all genes with $p > 10^{-5}$.

## Invariance to gene filtering

The gene filtering and mouse-human ortholog mapping steps considerably reduces the number of genes in the dataset. We computed ES weights *after* these gene filtering steps. All ES metrics except EP were invariant to the 'gene universe' of the dataset (all genes in the final dataset), consequently $ES_w$ are generally robust to gene filtering operations. However, $ES_{w*}$ (denoting a genes likelihood of being specifically expressed) is sensitive to the 'gene universe' as we use a null distribution pooled across genes to determine $ES_w$ significance (see the below section 'Determining the set of ES genes for each ES metric' for details).

## Choice of ES metrics

To implement a 'wisdom of the crowd' approach, we aimed at combining a diverse set of ES metrics. We choose ES metrics based on their documented evidence to identify tissue expression specificity based on bulk expression data (*Kryuchkova-Mostacci and Robinson-Rechavi, 2016*) or if they had been successfully applied on scRNA-seq datasets. Lastly, we aimed for metrics to estimate overexpression and not under-expression compared to other cell types.

Although we incorporated a t-test for differential expression as part of our ES metrics, we reasoned that test statistics or *P*-values from differential expression tests – for example DESeq2 (*Love et al., 2014*), MAST (*Finak et al., 2015*) – were not sufficient for making a useful ES metric. As a result, we did not choose to build $ES_\mu$ purely from DE test statistics. The result of DE tests was a list of genes ranked by the *uncertainty* or *signal-to-noise ratio* (*P*-value) of the estimate for difference in expression between cell type populations. For instance, a gene exhibiting a subtle difference in expression between two cell types and very low expression variance, would result in a highly significant DE P-value. A useful ES metric should not be biased towards assigning high expression specificity to identifying genes with low variance, in part because the variance of gene expression is

confounded by the cell type clustering resolution. Cell types with high heterogeneity will have higher variance of expression, resulting in downward-biased estimates of expression specificity for these cell populations when using a DE test as ES metric. In summary, our ES metrics do not seek to capture the uncertainty of difference in expression, but instead the relative magnitude of difference in expression between cell types.

## Overview of ES metrics used in this study

We here provide a short overview of ES metrics and their interpretation. In *Appendix 2—source data 1* we use $c$ to denote the focal cell type and $g$ to denote the gene for expression specificity estimation.

## ES metrics formula

For all formulas, we calculate $\mu_{g,c}$ as the average expression of gene $g$ in cell type $c$ in a dataset with $C$ cell types and $G$ genes.

## Gene Enrichment Score

GES is computed as the fold-change weighted by the fraction of cells expressing the gene. *Zeisel et al., 2018* showed that this measure was effective at identifying marker genes for cell types in the nervous system.

$$GES_{g,c} = \frac{\mu_{g,c}}{\mu_{g,\bar{c}}} \frac{f_{g,c}}{f_{g,\bar{c}}}$$

Here $\mu_{g,\bar{c}}$ is the average expression of all other cell types, $f_{g,c}$ is fraction of cells in cell type $c$ with non-zero expression of gene $g$, and $f_{g,\bar{c}}$ is the fraction of cells with non-zero expression in all other cell types.

## Expression Proportion

EP is calculated by dividing the expression of each gene in each cell type by the total expression of that gene in all cell types. Intuitively EP estimates the proportion of $g$'s total mean expression contributed by cell type $c$.

To remove the effect of differences in total expression between cell types, Skene et al., (2018) first normalize $\mu_{g,c}$ by the total expression of cell type:

$$\mu_{g,c}^* = \frac{\mu_{g,c}}{\sum_{g'}^G \mu_{g',c}}$$

EP is then calculated as:

$$EP_{g,c} = \frac{\mu_{g,c}^*}{\sum_{c'}^C \mu_{g,c'}^*}$$

We note that the first normalization step has no effect on our results, since we use common molecule count normalized expression data.

## Normalized Specificity Index

The Normalized Specificity Index (NSI) is modified from Specificity Index (*Dougherty et al., 2010*). Intuitively NSI estimates the average quantile (or relative rank) of gene $g$'s mean expression fold-change across all cell types. NSI is given by the formula:

$$NSI_{g,c} = \sum_{k \neq c}^K \frac{rank_g \left( \frac{\mu_{g,c} + \epsilon}{\mu_{g,k} + \epsilon} \right) - 1}{G - 1} / (k - 1)$$

Where $rank_g(\mu_{g,c} / \mu_{g,k})$ gives the position of gene $g$ in a descending-ordered list of 'fold-change' values for all genes, $\epsilon$ is a small numerical constant to prevent the fold-change from going to infinity as the denominator of the fold-change goes to zero.

We modified the SI metric from **Dougherty et al., 2010** for two reasons. Firstly, SI was originally developed for bulk gene expression data, which makes it less well-suited for count-based scRNA-seq, which is comprises an inflation of zero values. Secondly, SI values may take on arbitrary large values depending on the number of input genes, which makes the resulting SI values hard to interpret. We sought to normalize the SI scale to an intuitive [0–1] scale. Specially, we made three relevant modifications:

1. NSI is scaled by the number of genes to obtain a [0–1] scale.
2. NSI resolve ties using the minimum value instead of the average. This is relevant for sparse scRNA-seq where many genes will be tied for zero values.
3. NSI contains a small constant, $\epsilon$, to prevent SI from going to infinity as the denominator of the fold-change goes to zero.

We used the *specificity.index()* function of the pSI R package (v1.1, release 2014-01-30) as comparison for our modifications.

## Differential Expression T-statistic

We compute the t-statistic for gene $g$ as a measure of differential expression between the cell type $c$ and all other cell types.

$$DET_{g,c} = \frac{\mu_{g,c} - \mu_{g,\bar{c}}}{s_g\sqrt{1/n_c + 1/n_{\bar{c}}}}$$

Here, as above, $\mu_{g,\bar{c}}$ is the average expression of all other cell types, $n_c$ is the number of cells in cell type $c$, $n_{\bar{c}}$ is the number of cells in all other cell types, and $s_g$ is the pooled standard deviation estimate for gene $g$ given by:

$$s_g = \sqrt{\frac{\sum_{c'}^C (n_{c'} - 1)s_{c'}^2}{\sum_{c'}^C n_{c'} - 1}}$$

## Determining the set of ES genes for each ES metric

A key step in our approach is determining the set of *ES genes* for each ES metric. For each cell type we determined the set of specifically expressed genes, $G_s$, by testing the null hypothesis that a gene is more specific to a given cell type compared to cells selected at random. We compute empirical *P*-values of $ES_w$ by comparing observed weights to 'null' weights obtained by permuting the dataset's cell type annotations.

## Null distribution

We use an empirical null distribution because $ES_w$ for GES, NSI and EP do not have analytic statistical distributions to assess their significance. In addition, the analytical distribution for DET is not well-calibrated for (genome-wide) single-cell DE tests.

To construct our empirical null distribution we shuffled cells corresponding to the null hypothesis: gene X is equally specific to cell type A as it is to randomly selected cells. We constructed the null distribution such that our 'null cell types' had the same number of cells as the observed cell types. We note that in our null distribution, we kept the 'cell entities' and only the genes' average expression will change. Side remark: an alternative approach to constructing the null distribution, would be to shuffle genes corresponding to the null hypothesis that gene A is not more specific for cell type X than randomly selected genes. However, this null variant is not 'scale resistant' and will hence not work if genes are not on the same normalized scale.

## Nominal significance cut-off

We use an empirical *P*-value to find the cut-off between expression specific and non-expression specific genes. We use nominal significance ($p<0.05$) for this p-value (instead of an FDR adjusted cut-off) because we assume that our method is robust to inclusion of false positive ES genes.

## Normalization assumptions for computing $ES_{w*}$

For each cell type we calculate $ES_{w*}$, representing the genes' likelihood of being specifically expressed in a given cell type and for a given ES metric, by rank normalizing $ES_w$ for genes in $G_s$:

$$ES_{w*}(gene) = rank_g(ES_w(gene))/|G_s| \; if \; gene \in G_s$$

$$ES_{w*}(gene) = 0 \; if \; gene \notin G_s$$

We set $ES_{w*}$=0 for non-expression-specific genes (non-ES *genes*). This is important because we assume that we cannot meaningfully distinguish *between* non-ES genes, and hence they should all be given the same value.

Importantly, the rank normalization corresponds to the assumption that each cell type has a set of expression specific genes exhibiting a *linearly* increasing likelihood of being expression specific to the cell type. We apply this transformation to ensure $ES_{w*}$ have the same scale before combining into $ES_\mu$.

We note that this normalization strategy discards the 'magnitude' of the expression specificity (see below section 'Expression specificity profiles for ES metrics and average expression' for consequences of this). The linearity assumption essentially 'smooths' the non-linearity of $ES_w$ into a linear $ES_{w*}$ scale. We leave it for future work if other normalization strategies (e.g. inverse normal transformation or min/max) could offer an even better trade-off between dynamic range of expression specificity and robustness of the normalization.

## Expression specificity profiles for ES metrics and average expression

As discussed above, the magnitude of expression specificity is not captured in $ES_{w*}$ and consequently $ES_\mu$. To illustrate this, we plotted expression specificity profiles for two genes from the Mouse Nervous System dataset (***Appendix 2—figure 2***). The Agrp gene shows the same expression specificity profiles for ES metrics and log-transformed mean expression; only few cell types have high average expression levels (several order of magnitudes higher than the other cell types). The Pomc gene exhibits a slightly different expression specificity profile as it is more ubiquitously expressed. For both genes, the cell types with the highest average expression were also the top expression specific genes across most ES metrics.

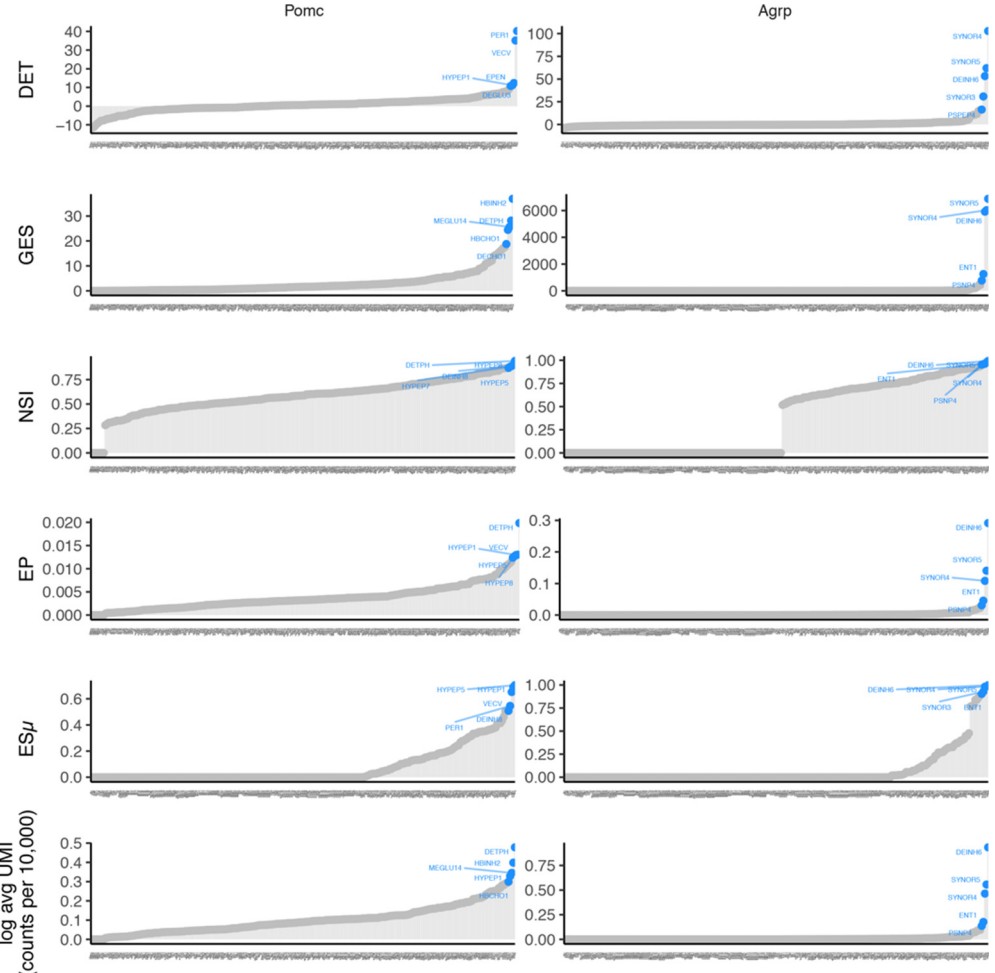

**Appendix 2—figure 2.** Expression specificity profile for *POMC* (left) and *AGRP* (right) AGRP is expressed in few cell types and *POMC* is expressed in slightly more cell types. Each panel row shows a different ES metric (normalized average expression is shown in the bottom row panels). The plot shows gene $ES_w$ (y-axis) for each cell type (x-axis, ordered by increasing values of $ES_w$). The five cell types with the largest $ES_w$ are highlighted. $ES_w$ values are estimated from the Mouse Nervous System dataset.

## Limitations of ES

Expression specificity is a relative measure as it depends on the background compendium of cell types included in the dataset. This means that we would obtain dramatically different $ES_\mu$ values for for example a neuronal cell type computed on the full Mouse Nervous System dataset and on a subset consisting only of the neuronal cell types. As a consequence, $ES_\mu$ values for similar cell types in two different datasets might be difficult to compare if they are computed on different background cell type compendia. Because expression specificity is inherently context dependent, there is a need for a 'common reference' dataset to ensure uniformity of values. We leave this for future work to explore.

## Future directions
### Preprocessing of ES

Gene expression imputation methods could be used to alleviate drop-out effects (missing values) causing downward bias of the cell type average expression estimates.

### Alternatives to rank $ES_{w*}$ normalization

We expect that other normalization strategies could offer a better trade-off between dynamic range of expression specificity and robustness of the normalization.

### Explore additional ES metrics

We find it unlikely that the four ES metrics used in this work captures all aspects of cell type expression specificity. It would be highly relevant to explore additional ES metrics.

### Context dependence of ES

We envision two strategies to combat the context dependency of ES values. One solution is to calculate a 'dataset diversity score' that measures diversity of the background compendium of cell types included in the dataset. This score could then potentially be taken into consideration when interpreting the results. A second solution could be including a 'common reference' dataset as background compendium of cell types. For example, use Seurat data integration methods (*Stuart et al., 2019*; *Butler et al., 2018*) to align the Tabula Muris dataset with the dataset of interest.

## Expression specificity robustness analysis

### Aim

ES is inherently dependent on the cell type composition of the dataset. Still, the $ES_{w*}$ should primarily reflect the properties of the cell type and not the context of the dataset. For example, we wish to obtain similar $ES_{w*}$ when we replicate a scRNA-seq experiment even if the cell type composition has shifted slightly. In other words, it is desirable for an ES metric to produce similar $ES_{w*}$ in varying contexts of cell type composition. We here aim at assess this characteristic: the *robustness* of ES metrics. A robust metric will yield similar results in changing cell type contexts.

### Methods

To assess the robustness of ES metrics, we defined a Robustness Score (RS), which measures the ability of an ES metric to reproduce $ES_{w*}$ in changing cell type computations. We used $ES_{w*}$-baseline to denote $ES_{w*}$ of a selected focal cell type computed on the full dataset, that is all cell types. Further, $ES_{w*}$-subset denotes $ES_{w*}$ of the focal cell type computed on a random subset of the data. For a given focal cell type and data-subset-proportion, we performed the following procedure:

1. Create a subset of the dataset by randomly sampling the specified subset proportion (e.g. 20%) of cell types from the full dataset. The focal cell type is excluded during sampling.
2. Add the focal cell type to the sub-sampled dataset.
3. Compute $ES_{w*}$ for the focal cell type to obtain $ES_{w*}$-subset.
4. Compute Pearson's correlation coefficients between $ES_{w*}$–baseline and $ES_{w*}$–subset to obtain RS-subset.

The sampling procedure was repeated 100 times for each ES metric and data subset proportion. When computing *RS-subset*, we adjusted for zero-inflation by removing genes with zero values in both $ES_{w*}$-baseline and $ES_{w*}$-subset. We reported RS values averaged over the 100 repetitions.

We performed the above procedure using 12 distinct *Mouse Nervous System* cell types as focal cell types (ABC, ACNT1, ACTE1, COP1, DEINH3, EPMB, EPSC, MGL1, OPC, PVM1, TEGLU1, VLMC1). To ensure the generalizability of our results, the 12 focal cell types were selected as representatives of each major cell type class (astrocyte, ependymal, glia, neuronal, oligodendrocyte and vascular cells). We summarized the results of each subset proportion across the 12 focal cell types by computing the mean and standard deviation of the mean RS across cell types.

### Results

We observed that $ES_{\mu}$ achieved the highest mean RS with low variation across all tested focal cell types (*Appendix 2—figure 3*). GES exhibited the lowest mean RS with high variation across focal cell types. The mean RS across all ES metrics was generally high (>0.8) for subsets greater than 10%

of all cell types. All ES metrics exhibited similar mean RS for subsets greater than 50%. In summary, $ES_\mu$ is the most robust ES metric to changes in cell type composition.

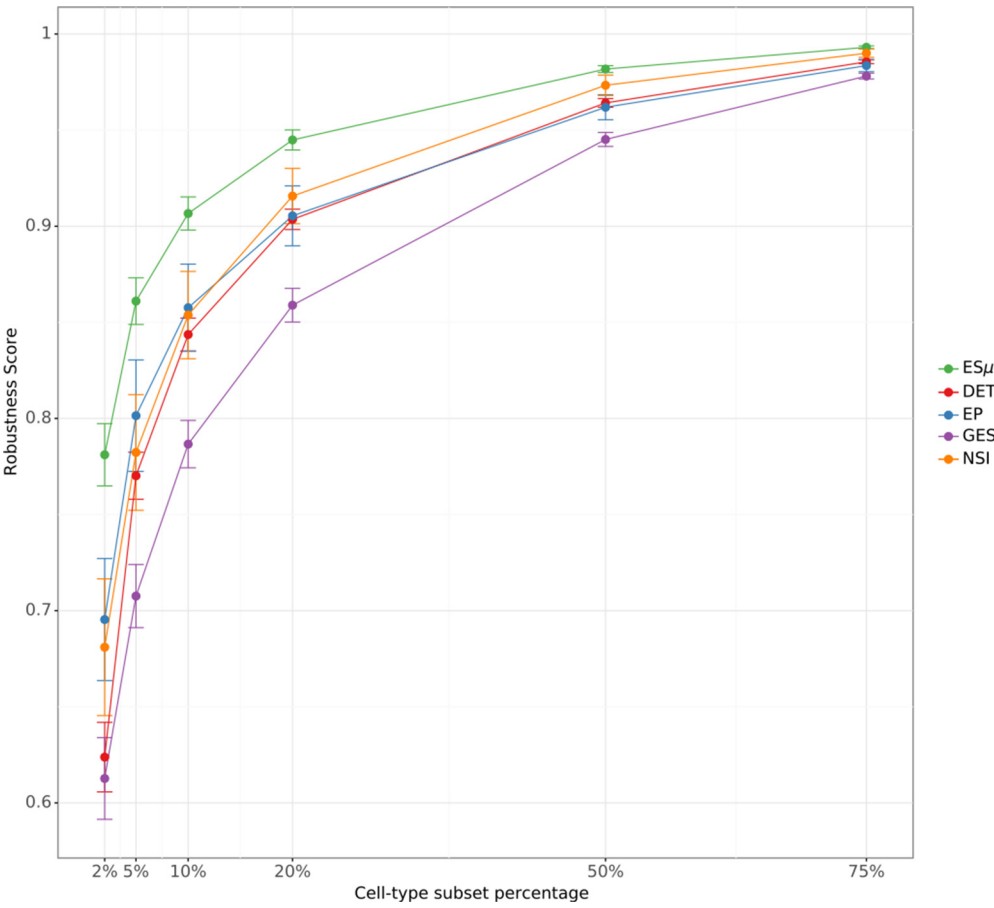

**Appendix 2—figure 3.** Robustness of ES metrics. The figure depicts the robustness score (RS) for each ES metric as a function of cell type subset percentage. Each point represents the mean RS across 12 focal cell types. Error bars indicate standard error of the mean.

## Appendix 3

### Cell type gene co-expression networks

Introduction

Here we provide detailed methods for the gene module analysis and its limitations.

## Gene module analysis

We identified cell type gene co-expression networks using a modified version of the robust WGCNA (rWGCNA) framework proposed by *Gandal et al., 2018*. To identify gene co-expression networks (*gene modules*) operating within a cell type, the input to rWGCNA was expression data for individual cell types. Our framework consists of the following steps:

## Pre-processing

We used the *Seurat* R package (version 2.3). We filtered out genes expressed in fewer than 20 cells and removed any cell clusters containing fewer than 50 cells. The *NormalizeData()* function was used for normalizing raw expression values to a common transcript count (with n = 10,000 transcripts as a scaling factor), log-transformation (log(x+1)), before scaling and centering with the *ScaleData()* command to arrive at a matrix of Z-scores. Principal component analysis was carried out using the *RunPCA()* function to find 120 Principal Components (PCs). Genes were then ranked by their highest absolute loading value on any given PC and the top 5000 genes within each cell cluster were selected for co-expression analysis. We mapped mouse genes to orthologous human genes using Ensembl (v. 91), keeping only 1–1 mapping orthologs as done for the ES calcuations.

## Robust weighted gene correlation network analysis and adjacency matrix calculation

We used the *WGCNA* R package (version 1.66). For computing the gene-gene adjacency matrix we used the Pearson correlation coefficient and the *signed hybrid network* parameter. The *pickSoftThreshold()* command was used to identify soft thresholding powers. Powers corresponding to the top 95[th] percentile of network connectivity or above were discarded and the lowest soft threshold power between 1 and 30 to achieve a scale free topology R squared fit of 0.93 was selected; if no powers reached 0.93, the thresholding power with the highest R squared was chosen instead.

## rWGCNA consensus topological overlap matrix calculation

The expression data was resampled as described in *Gandal et al., 2018*; drawing two thirds of the cells at random without replacement 100 times. The consensusTOM() function was used with a consensusQuantile of 0.2 to compute a signed consensus *topological overlap matrix* (TOM), and genes were then filtered using the output of the *goodGenesMS()* function called by consensusTOM().

## rWGCNA hierarchical clustering

The consensus TOM matrices were converted to distance matrices and the *hclust()* function was used with the *average* method to cluster genes hierarchically. The *cutreeHybrid()* function was used with a *deepSplit* of 2, *minClusterSize* of 15 and *pamStage* set to TRUE to carve the dendrogram into modules. The *mergeCloseModules()* function was used to compute module eigengenes, the vector of cell embeddings on the first principal component of each module's expression submatrix. The same function was used to merge modules, using a *cutHeight* of 0.15 or less, corresponding to a Pearson correlation between module eigengenes of 0.85 or greater.

## rWGCNA gene-module connectivity

The module eigengenes and expression matrices were used with the *signedKME()* function to compute gene-module Pearson correlations, or *kMEs*, a measure of how close each gene is to each

module. To ensure tightly connected modules, genes whose correlations with their assigned modules eigengene was not statistically significant after correcting for multiple testing, using the Benjamini-Hochberg false discovery rate (FDR) method, were removed.

## Limitations of identifying gene modules from scRNA-seq data

In this section we discuss the relevance and limitations of identifying gene co-expression networks on a by-cell type basis. Some of these limitations have also been discussed elsewhere (*Chen and Mar, 2018*; *Crow and Gillis, 2018*) and may explain the lack of identified BMI prioritized cell type gene co-expression networks. While WGCNA has primarily been applied to bulk RNA-seq expression data, several recent studies have demonstrated its effectiveness in single-cell data (*Nowakowski et al., 2017*; *Tasic et al., 2016*). As gene-gene correlations reflect gene expression variability, WGCNA analysis of heterogeneous bulk expression data consisting of several distinct cell type populations, will result in a coarse set of genes modules, largely reflecting cell type heterogeneity. WGCNA analysis of pure cell type populations from scRNA-seq data will result in more specific genes modules. However, constructing gene networks on pure cell type populations poses another problem: the true gene-gene correlations are more difficult to estimate in expression data with limited gene expression heterogeneity (expression variability). In addition, scRNA-seq data have increased technical noise (dropout effects) compared to bulk RNA-seq data, which reduces the ability to estimate the true gene-gene correlations.

## Appendix 4

### Heritability of prioritized BMI cell types

Heritability of prioritized BMI cell types

In CELLECT we utilized a continuous representation of cell type expression, assuming there is a positive relationship between genes' ES values and their importance for a given trait. Stratifying genes into quintiles based on their ES values for a given cell type and calculating the enrichment of BMI heritability for each stratum, showed that an increase in ES was reflected in an increase in BMI heritability for the BMI-prioritized cell types (*Appendix 4—figure 1a*; first row). Other well-powered UKBB anthropometric traits GWAS did not exhibit any relationship between ES quintile and trait heritability for the BMI-prioritized cell types (*Appendix 4—figure 1a*; second and third rows). These results indicate, that our models are well calibrated. To better understand the proportion of BMI heritability explained by variants mapped to a certain cell type, we compared the proportion of BMI heritability explained by each of the 10 cell types to the proportion trait-heritability explained by cell types known to play a key role in the given trait or disease. We found that the most enriched cell type for BMI, was the cortical TEGLU4 cell type, which comprised specifically expressed genes with genetic variants accounting for 28.5% of the SNP heritability (h2g) for BMI (*Appendix 4—figure 1b*). Similarly, we estimated that SNPs mapped to genes specifically expressed in the pancreatic beta cells, an insulin secreting cell type playing a pivotal role in type 2 diabetes, explained 27.8% of the heritability for type 2 diabetes; hepatocytes explained 27.1% in the heritability for low-density lipoprotein (*Teslovich et al., 2010*); T cells explained 20.7% of the heritability for rheumatoid arthritis (*Okada et al., 2014*); and, finally, mesenchymal stem cells explained 20.4% of the heritability for human height (*Loh et al., 2018*; *Appendix 4—figure 1b*). These results show that genetic susceptibility to obesity conferred by variants mapped to genes specifically expressed in the cortical cell type roughly corresponds to the heritability conferred by pancreatic beta cell variants on type 2 diabetes.

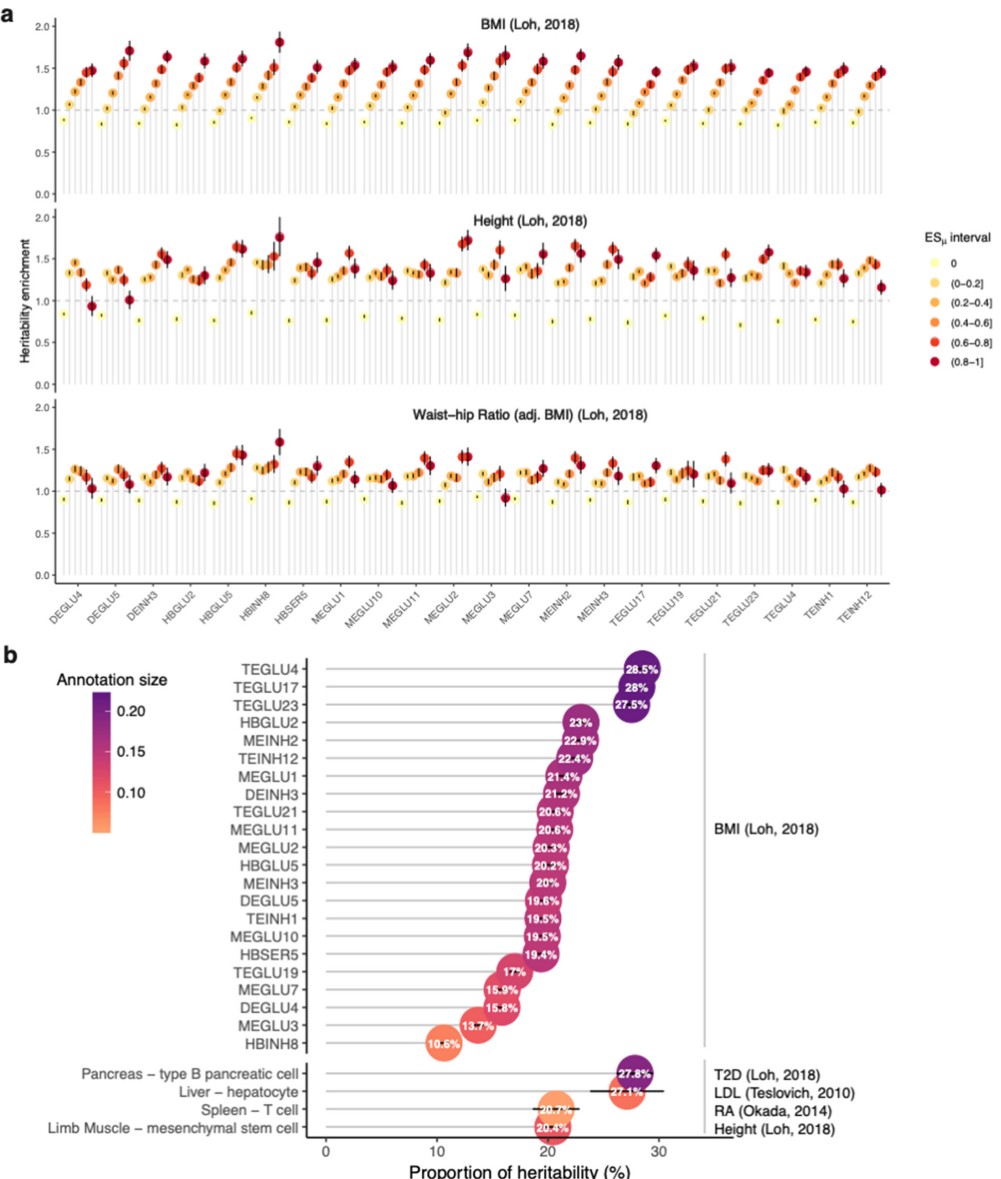

**Appendix 4—figure 1.** Heritability of BMI prioritized cell types. (**a**) Heritability enrichment of cell type $ES_\mu$ intervals. Heritability enrichment was estimated using S-LDSC on cell type $ES_\mu$ annotations partitioned into five equally spaced intervals and an interval including $ES_\mu=0$. The intervals 0 and (0.8–1) represent the heritability enrichment of by the variants with the lowest and highest $ES_\mu$ values, respectively. Error bars represent 95% confidence intervals. The top, middle and bottom panel show results for BMI, height and waist-hip ratio, respectively. BMI heritability enrichment increases with increasing $ES_\mu$ value for prioritized cell types. (**b**) Proportion of BMI heritability explained by prioritized cell types. We used S-LDSC to estimate the proportion of trait SNP heritability explained by each cell type annotation. For comparison we report the proportion of heritability explained by cell types with known etiology for selected traits: type 2 diabetes (T2D), low-density lipoprotein (LDL), rheumatoid arthritis (RA) and human height. Circles are colored by annotation size reflecting the proportion of variants covered by the cell type annotation (a value of one means that all variants were covered). Error bars represent 95% confidence intervals.

