## [Decision Letter]

Thank you for submitting your article "Mapping heritability of obesity by brain cell types" for consideration by *eLife*. Your article has been reviewed by three peer reviewers, and the evaluation has been overseen by a Reviewing Editor and Naama Barkai as the Senior Editor. The reviewers have opted to remain anonymous.

The reviewers have discussed the reviews with one another and I have drafted this summary to help you prepare a revised submission. As you can see, the concerns are substantial and will need to be addressed, before we can make a final decision. We typically allow two months for revisions, but given the COVID-19, we understand that activities in your lab may have slowed down or even cancelled. Therefore, we are happy to extend this timeframe, if needed.

The two main concerns can be summarized as follows (please, also find their specific concerns below):

For these tool kits to be useful they should be able to identify tissues, cell types and/or genes that are well-established for a given disease. We feel that the validation of the tool kits, at least for obesity, is not convincing at the moment. For example, in the context of obesity, it would seem that the tool kits would also identify PVH and ARH neuronal cell types, given that MC4R and POMC are among the GWAS-identified obesity loci.

Another concern, possibly related to the first concern, is the quality of data used. We believe that the currently used data may not have sufficient sequencing depth to identify all relevant cell types and that differences in the representation of cell types in brain regions that are very heterogeneous, such as the hypothalamus, may impact the results. It will be important to discuss how quality of data impacts results.

Taken together, we need to be convinced that the tool kits generate robust findings. However, this is currently hard to assess as the quality of the data used for this proof-of-principle does not seem great. Therefore, it will be important to discuss how the quality of the data may impact the results generated by the tool kits, ideally by using high-quality data relevant for the disease used in the proof-of-principle example (i.e. obesity).

Reviewer #1:

Thimshel et al. have developed two complimentary computational toolkits, CELLEX and CELLECT, to integrate single cell RNA sequencing data with GWAS data to prioritize cell types that are key in disease. It is based on the assumption that genes that cause disease are expressed in tissues and cell types that are key to the disease. In the past, the authors have shown with tissue enrichment analyses that the brain plays a key role in obesity, consistent with findings from monogenic forms of obesity. With these new pipelines, the aim to target the cell types involved. CELLEX combined 4 gene expression specificity measures into one score to indicate in which cell type a given gene is specifically expressed. CELLECT quantifies the enrichment of heritability in/near genes specifically expressed in certain cell types. By applying CELLEX and CELLECT to genes prioritized from GWAS for BMI, they identified 26 neuronal cell types across the brain, rather than being restricted to a limited number of tissues.

CELLECT depends on other prioritization software, such as S-LDSC, MAGMA, DEPICT, etc. For the current analyses, the authors used S-LDSC. How confident can we be that the genes prioritized are indeed the causal genes and what would the results look like if DEPICT (the authors' own software), or MAGMA were used? At least a justification for using S-LDSC needs to be given.

The assumption for CELLECT is that genes that have a high expression specificity (e.g. in the brain) are more important than genes that are widely expressed (or not expressed much) across cell types and tissues. How confident can one be that this is a correct assumption (across the board)? More specifically, are genes that are widely expressed or not expressed or expressed in other tissues not important in obesity? For example, LEP is predominantly expressed in adipose tissue, but is known to signal to the brain to influence food intake.

While there is indeed growing evidence that not only the hypothalamus plays a key role in obesity, one would expect that well-established obesity genes, tissues and cell types would be identified by new methods.

According to the Tabula Muris Nature paper (2018), it includes cerebellum, cortex, hippocampus and striatum. Can cell types be distinguished by these tissues, in particular given that the hippocampus was the most enriched tissue in Locke et al., one may expect some more specific evidence for this tissue/cell type.

It was stated that coding mutations in "syndromic" forms of obesity were chosen for replication of methods. It should be noted that this list includes genes identified for monogenic and syndromic forms of obesity– it seems the term "syndromic" is not used correctly (ie. include monogenic and syndromic). While not unimportant, it seems that mutations in monogenic forms of early-onset extreme obesity are more relevant for common forms of obesity than syndromic. Therefore, these analyses may be better when stratified.

Reviewer #2:

More than 250 genetic loci have been implicated in human obesity. While others have shown that the GWAS loci are disproportionately expressed in the brain, the key cell types affected is not known. The authors developed a computational pipeline that leverages the growing body of single cell RNA-seq data to identify specific cell types that are likely to be preferentially impacted by the GWAS variants. The authors developed two computational tools that are released as open-source packages for Python programming languages: CELLEX and CELLECT. Application of CELLEX (Cell type EXpression-specificity) to scRNA-seq data integrates four metrics of expression specificity into a single parameter (Expression Specificity, ES) to identify genes that are preferentially expressed in a particular cell type. CELLECT (Cell type Expression specific integration for Complex Traits) integrates the information from CELLEX with GWAS data to identify cell types that have enriched expression of nearby genes, and thus are likely to contribute to the pathophysiology of obesity. They validated the CELLECT tool by showing that it could identify relevant cell types for 10 different GWAS databases (i.e. neurons in the case of the BMI GWAS). Running this analytical pipeline on 256 from the mouse nervous system scRNA-seq and BMI GWAS databases identified 22 enriched cell types in 8 brain regions. Surprisingly, none of these cell types were hypothalamic, which could be explained by sparse sampling of the large number different hypothalamic cell types in the scRNA-seq datasets. Direct interrogation of 347 hypothalamic cell types identified 4 enriched cell types in the VMH, LHA and POA. If their assumptions and models work as predicted, these tools would significantly advance efforts to uncover cell types and circuits regulating susceptibility to obesity, or any other complex trait of interest. While any computational toolkit has its limitations, several issues must be addressed in order to evaluate the reliability of the data generated with this pipeline.

1) The authors need to explain why PVH and ARH neuronal cell types were not identified in their analyses. POMC and MC4R are GWAS loci, and mutations in these genes produce severe obesity in humans and in mouse models. CELLEX/CELLECT implicated a cortical cell type in mediating the effects of mutations in MC4R and the ciliary gene ADCY3. Genetic studies in mice provide strong evidence that disruptions of MC4R or cilia formation in the PVH is sufficient to cause obesity.

(1a) To what degree are the analyses impacted by differences in the representation of cell types in brain regions that are very heterogeneous? Are ES scores inflated in brain regions where the cell types are more homogeneous or better annotated? To what extent will this issue be mitigated as more scRNA-seq datasets are published in heterogeneous tissue such as the hypothalamus?

(1b) The implication of a cortical cell type in mediating influences of MC4R and ADY3 is novel and unexpected. Demonstration that disruption of MC4R or cilia formation in the cortex would go a long way to allay concerns about the validity of the CELLEX/CELLECT toolkit.

Reviewer #3:

The authors report the development of in silico tools that aspire to help identify cell types that participate in body weight regulation by aligning BMI GWAS loci with scRNA-seq datasets spanning an array of tissues. The key assumption for this analysis is that "in order for a disease to manifest in a given tissue or cell type the set of disease-causing genes must be active and expressed in the given tissue or cell type. In other words, the model presupposes that high/increased expression (and not decreased/lack-of expression) of a gene results in disease." The wording states poorly this important assumption. The authors assume that in order for a gene to be important in a specific tissue, it needs to be expressed. It has nothing to do with the change in expression of a certain gene per SNP allele. If the gene is not expressed in a certain cell type, it is unlikely that the presence of one allele or the other influences the disease or trait.

1) In addition to the narrow field of cell state (adult versus development) that Timshel et al. point out, two important limitations/confounds are not highlighted: a) scRNA-seq depth of sequencing limits the detection of lowly expressed upstream signaling components critical to body weight regulation and b) gene prioritization is based on a variety of criteria that vary from one GWAS study to the other. Calling the wrong gene would affect the input in the present analysis.

2) Given all these limitations, a proof of principle example should be tested. Mc4r is the perfect candidate, alas the absence of scRNA-seq data from highly relevant cell types including the PVH. The fact that hypothalamic BMI GWAS enrichment is low runs against the validation of the tool. So does the leptin receptor result, possibly indicative of the limitation of scRNA-seq transcript capture.

3) The authors state that CELLEX is currently not set up for adjustment of unwanted sources of variation such as batch effects. The authors should clarify the magnitude of variation that is tolerated without affecting the ES metrics.

4) Furthermore, given this limitation of CELLEX, how were the different datasets from the hypothalamus combined? Was each dataset run separately through CELLEX or were the 347 cell types combined to a CELLEX input? Does CELLEX take count data as input or does it work with other measures such as TPM?

5) The use of the WGCNA method does not seem to serve a purpose. If the authors need to undertake network analysis, they should utilize methods developed for scRNA-seq or completely omit this from the manuscript.

[Editors' note: further revisions were suggested prior to acceptance, as described below.]

Thank you for submitting your article "Genetic mapping of etiologic brain cell types for obesity" for consideration by *eLife*. Your article has been reviewed by three peer reviewers, and the evaluation has been overseen by a Reviewing Editor and Naama Barkai as the Senior Editor. The reviewers have opted to remain anonymous.

The reviewers have discussed the reviews with one another and the Reviewing Editor has drafted this decision to help you prepare a revised submission.

The reviewers agreed that the paper has improved substantially and the authors have addressed their concerns. Nevertheless, there are some remaining concerns of which we'd like you to address the following [no need to address the individual reviewer comments at this point];

– Both reviewer 1 and 2 would like you to put your results/observations into context; what does one do next with the "identified" genes and tissues.

– Reviewer 2 would like to make sure your tool does readily identify the genes driving the link a specific tissue/cell type.

While experimental validation would be ideal, as suggested be reviewer 3, we don't expect you to do this within the context of the current submission.

For your information; below are the individual reviewer comments.

Reviewer #1:

The authors have been very thorough in addressing my (and other reviewers') concerns.

What's left is putting the generated findings in context; i.e. would you suggest to jump straight to functional follow up, or should researchers further validate observation, and if so, how would you suggest they do this.

Reviewer #2:

If they work as advertised, the analytical tools presented here would permit the unbiased identification of novel neural substrates of genetic influences on complex disease traits. While this information is very valuable if accurate, it also has the potential to lead investigators down a wild goose chase that would waste valuable resources. In their revised manuscript, the authors address concerns about the conspicuous absence of ARH POMC and PVH MC4R neurons from the list of cell types that are likely to be preferentially impacted by the BMI GWAS variants. They add a deeper analysis of existing datasets and are now able to detect a signal in a subset of POMC neurons. The lack of published PVH datasets is an obstacle to performing similar analyses to detect MC4R neurons.

The major question remaining is related to the reliability of the unexpected (and potentially exciting) targets identified. For example, analysis of schizophrenia and intelligence GWAS loci revealed a linkage with some pancreatic cell types and analysis of height loci revealed a linkage with tracheal cell types (Figure 2B). In the context of obesity GWAS loci, the strongest enrichment is in the cerebellum, cortex and amygdala (Figure 5C). For these tools to be valuable to biologists (beyond providing an easy way to generate another figure in a paper), it is critical that they provide a path forward to validating these unexpected associations. At a minimum, the tools should readily identify the specific genes that are driving the linkage to a specific cell type (for example, see the Enrichr platform https://amp.pharm.mssm.edu/Enrichr/). This information would permit biologists to design experiments to investigate the novel relationships identified here.

Reviewer #3:

Lack of "proof of principle" remains a major concern. Assertions to the effect that certain known players such as Pomc do not regulate body weight in adult neurons is questionable (Mol Endocrinol. July 1, 2013; 27(7): 1091-1102). Experimental proof of at least one of these assertions is critical for the demonstration of validity and physiological relevance. As is, it is hard to discern whether these findings were a consequence of methodology rather than biology.

---

## [Author Response]

The two main concerns can be summarized as follows (please, also find their specific concerns below):For these tool kits to be useful they should be able to identify tissues, cell types and/or genes that are well-established for a given disease. We feel that the validation of the tool kits, at least for obesity, is not convincing at the moment. For example, in the context of obesity, it would seem that the tool kits would also identify PVH and ARH neuronal cell types, given that MC4R and POMC are among the GWAS-identified obesity loci.

We completely agree that Mc4r neurons from the paraventricular hypothalamus (PVH) and Pomc neurons from the arcuate nucleus of the hypothalamus (ARC) undoubtedly play a key role in human obesity. That being said, we have some reservations towards constructing a PVH single-cell validation dataset and about being too focused on using current ARC datasets and especially current Pomc populations as positive controls.

A single-cell atlas for the PVH yet needs to be published and, unfortunately, current datasets of the hypothalamus do not contain any PVH Mc4r^+^ populations. Campbell et al.(1) refer to one subtype enriching for Mc4r neurons namely *n19.Gpr50*; we found that Mc4r was specifically expressed in these neurons (ESμ=0.99) and that CELLECT exhibit nominal significance (*P=0.04).* However, this population likely resides in the ARC and may thus not represent a good robust positive control for canonical Mc4r^+^ neurons. Constructing a single-cell atlas of the PVH, which is a very heterogeneous area (2), constitutes a considerable effort both in terms of time and costs, and would in our view merit a publication in itself. We added the following sentence to the Discussion: “Second, the datasets used in this work should not be regarded as complete atlases because they are likely to miss relevant cell types such as Mc4r-positive neurons, which are known to play a key role in obesity”.

In the following we will discuss using current Pomc^+^ populations as positive controls. Despite substantial insights into the role of Pomc in energy regulation, insights into how it impacts predisposition to human obesity remain incomplete. For example, the exact timing during which Pomc exerts its effect on genetic susceptibility remains to be understood in greater detail. Kehra et al. showed that genetic variants associated with body mass index (BMI) start exerting their effect during early childhood and early adolescence which roughly corresponds to early postnatal development in mice (3). Recently, van der Klaauw et al. showed that coding variants associated with extreme obesity enrich for semaphorin genes regulating Pomc maturation, a process taking place during that postnatal developmental time period in mice (4). Consequently, because current hypothalamic transcriptomics data are based on hypothalami from adult mice, one has to be cautious when relying on them as a possible control.

Despite the above-mentioned limitations, we do agree that the paucity of hypothalamic signals needs to be investigated further. Towards that end we performed four additional analyses:

We first focused on confirming that the current hypothalamic single-cell data actually allow for detection of known obesity risk genes. Towards that we, we (a) identified Campbell et al. ARC neuronal subpopulations in which the “high-confidence obesity genes” (from studies of monogenic- and extreme obesity and genes with protein-altering variants associated with obesity) were expressed, and then (b) plotted their expression specificity. Pomc, Lepr and Mc4r were detected in the relevant neuronal cell populations (please see new panel b in Figure 5) and correctly identified by CELLEX as being specifically expressed in these cell types. Among the 23 high-confidence obesity genes, 20 were part of the ARC dataset (dropouts: Lep, Rapgef3, Znf169) and 18 of them had detectable expression levels (dropouts: Wnt10b and Znf169) and 17 were expressed in at least 10% of the cells of one cell type and specifically expressed in at least one neuronal ARC cell type (dropout: Gipr; Figure 5—source data 5). (The Gipr gene is part of the G-protein coupled receptor gene family, a class of genes typically lowly expressed and thus difficult to identify in current single-cell RNA-seq data.) These results indicate that lack of significant BMI GWAS enrichments for hypothalamic neurons is unlikely to be driven by a missing ability to detect “core” obesity genes but rather can be explained by a lack of polygenic BMI GWAS signal in the other genes in these cell types (e.g. a lack of expression of semaphorins). We made the following changes to the manuscript:

Updated these findings in the Results section (subsection “Ventromedial hypothalamic Sf1- and Cckbr-expressing cells enrich for BMI GWAS”).Added Figure 5B.Added Figure 5—figure supplement 1.

To confirm that current hypothalamic datasets and CELLEX enable detection of co-specifically expressed within relevant cell types, we tested whether any of the ARC neuronal cell populations enriched for the 23 high-confidence obesity genes. Among four significant cell populations, the top hit was one of the Pomc^+^ populations. This finding indicates that key hypothalamic cell populations co-express relevant obesity genes and that the CELLEX methodology enables detection of relevant hypothalamic cell types (In the CELLECT results, four of the five Pomc^+^ cell populations were nominally enriched.) Together these results suggest that while there is a significant overlap between genes implicated through studies of monogenic obesity and studies of common variants associated with BMI, there are important differences which remain to be understood.

Apart from adding these observations to the Results, we added the following sentence to the Discussion: “We show that while the polygenic enrichment signal is highly correlated with enrichment of high-confidence obesity genes, this alignment diverges for hypothalamic neuron populations (including Pomc-positive neurons) suggesting that common genetic susceptibility to obesity acts on a more broadly distributed set of neuronal circuits across the brain”.Added Figure 5—source data 6 showing results for the high-confidence obesity geneset enrichment across all ARC cell populations.

We next tested whether there in general was an overlap in cell types enriching for relevant obesity genes and the cell types prioritized for BMI by CELLECT. Towards that end, we leveraged the set of high-confidence obesity genes to compute enrichments across cell types from all hypothalamic studies and correlated these cell type-specific enrichments with the results obtained from CELLECT. The correlations averaged at Pearson’s rho=0.40 and were particularly high for the ARC (Pearson’s rho=0.5, *P*=2.2x10^-5^) and LHA (Pearson’s rho=0.6, *P*=9.1x10^-10^) datasets. These results confirm that, overall, the CELLEX and CELLECT toolkits are able to identify relevant hypothalamic cell types.

We added a section to the Results describing these results (subsection “Ventromedial hypothalamic Sf1- and Cckbr-expressing cells enrich for BMI GWAS”).Updated Figure 5—source data 4, which is now omitting the syndromic obesity genes (see below reviewer comment).Added Figure 5—source data 7 showing the Pearson’s correlations for all hypothalamic single-cell datasets.

Finally, previous studies have suggested that the hypothalamus is not necessarily the most BMI-GWAS enriched tissue(5). To relate the enrichment of BMI heritability in genes specifically expressed in adult human hypothalami compared to genes specifically expressed in other human brain areas, we applied CELLEX and CELLECT on the most recent Genotype Tissue Expression (GTEx) Consortium human post-mortem gene expression data. Analysis of a total of 16,027 RNA-seq samples (from 945 individuals) revealed that the hippocampus and several other brain areas exhibited stronger enrichment signal than the hypothalamus. In contrast, the high-confidence obesity genes enriched most strongly for the hypothalamus (*P*=3.9x10^-4^, FDR<0.05). These results support our previous observation that despite overlaps, core obesity genes and polygenic signal point to slightly different parts of the brain. We added the following parts to the manuscript:

A description of the GTEx findings to the Results (subsection “Ventromedial hypothalamic Sf1- and Cckbr-expressing cells enrich for BMI GWAS”).Panel c to Figure 5 showing the GTEx results.Added Figure 5—source data 10 – 12 containing the GTEx tissue annotations, GTEx CELLECT results and GTEX high-confidence obesity genes enrichments results.

All together these observations suggest that current hypothalamic scRNA-seq data when analyzed with CELLEX and CELLECT can identify relevant hypothalamic genes and cell types. They furthermore suggest that (a) the polygenic susceptibility underlying obesity is likely to be distributed across several cell types and brain regions, and (b) concurrent use of polygenic and core signal will provide relevant insights into the biology of obesity. However, to acknowledge the overall concern about the lack of signal for Pomc and Mc4r^+^ neurons we clarified the corresponding sentence in the Discussion: “First, the scRNA-seq data analyzed here were derived from late postnatal, adult and predominantly wildtype mice; future work is needed to assess the role of Pomc+, Agrp+ and Mc4r+ and other hypothalamic cell types during developmental stages and relevant obesogenic perturbations in human obesity”.

Another concern, possibly related to the first concern, is the quality of data used. We believe that the currently used data may not have sufficient sequencing depth to identify all relevant cell types and that differences in the representation of cell types in brain regions that are very heterogeneous, such as the hypothalamus, may impact the results. It will be important to discuss how quality of data impacts results.

We agree that this is an important point. To make sure that our analyses are not confounded by sequencing depth and other technical factors, we constructed 1,000 GWAS summary statistics based on simulated Gaussian phenotypes with no genetic basis and used them to assess the impact of possible confounders on CELLECT results. Running CELLECT on these GWAS, we did not find any correlation with the number of genes detected for a given cell type (i.e. sequencing depth) nor the number of (unique) transcripts measured for a given cell type. We found a negligible correlation with the number of cells covering a given cell type (Pearson’s rho=0.01, *P*=4.0x10^-4^), which disappeared when we adjusted for the number of specifically expressed genes for a given cell type, suggesting cell types with few cells may deflate CELLECT enrichments.

In the Results we have added that “Finally, using 1,000 “null GWAS” constructed based simulated Gaussian phenotypes with no genetic basis we found that CELLECT had a properly controlled type 1 error and that results were not confounded by the median number of genes and transcripts per cell (there was a negligible correlation with the number of cells for a given cell type [Pearson’s rho=0.01, p=4.0×10^–4^], which disappeared when we adjusted for the number of ESμ genes for a given cell population)”.Added Figure 2—figure supplement 1 illustrating the above results.

We completely agree that the current atlases do not represent the final complete maps of all cell types and states in the brain, and that ongoing and future efforts will identify additional transcriptional states potentially relevant to obesity. Regarding the representation of cell types in heterogeneous brain regions (also brought up by reviewer #2, major concern #1A); this is a very relevant question that we considered in length while developing CELLEX and CELLECT. It is correct that the ESμ score is a relative measure as it, for a given cell population, depends on the other cell types contained in the given dataset. All the single-cell data sets we analyzed here cover broad cell type categories (cell types from peripheral tissues or neuronal and glia cell populations). ESμ values decrease when reducing cell population heterogeneity. For instance, we found that ESμ values became less specific when running CELLEX on ARC neurons only, compared to constructing ESμ values based on all ARC cell types: among the 18 high-confidence obesity genes detected in the ARC dataset, 16 exhibited decreased ESμ values when only analyzing neuronal cell types. These results suggest that more detailed regional atlases will add variation that should help to identify genes that are specifically expressed in relevant cell types and under relevant cell states.

Results: “Moreover, we observed that ESμ values increased when increasing cell population heterogeneity; 16 out of the 18 ARCME-detected high-confidence obesity genes became more specifically expressed when running CELLEX on all ARC cells compared to ARCME neurons only. Together these results indicate that (a) current hypothalamic single-cell data and our CELLEX methodology are sufficient to detect relevant cell populations and that upcoming regional atlases with increased cellular heterogeneity will allow for discovery of additional relevant cell populations and cell states, […]”.Added Figure 5—source data 9 to support the above-mentioned results.

Taken together, we need to be convinced that the tool kits generate robust findings. However, this is currently hard to assess as the quality of the data used for this proof-of-principle does not seem great. Therefore, it will be important to discuss how the quality of the data may impact the results generated by the tool kits, ideally by using high-quality data relevant for the disease used in the proof-of-principle example (i.e. obesity).

We acknowledge the concern regarding the lack of significant signal and have done our best to create a more cohesive context within which to understand the hypothalamus results, while acknowledging gaps in our knowledge and emphasizing the importance of future work in this area. Basically, in our manuscript we are now reporting three lines of evidence showing that our two toolkits provide robust findings. First, CELLECT is able to identify relevant cell types for traits with a slightly better known etiology (e.g. hepatocytes for triglycerides and low-density lipoprotein and mesenchymal stem cells for human height). Second, for obesity we now show that the CELLECT results are highly correlated with the cell type enrichment derived based on high-confidence obesity genes, and that analysis of GTEx samples results in similar rankings of the hypothalamus. Finally, we show that the type-1-error rate is well calibrated and that the results are not driven by sequencing depth. All in all, we are strongly convinced that our CELLECT results provide useful new insights into brain cell types likely mediating susceptibility to common obesity.

Reviewer #1:Thimshel et al. have developed two complimentary computational toolkits, CELLEX and CELLECT, to integrate single cell RNA sequencing data with GWAS data to prioritize cell types that are key in disease. It is based on the assumption that genes that cause disease are expressed in tissues and cell types that are key to the disease. In the past, the authors have shown with tissue enrichment analyses that the brain plays a key role in obesity, consistent with findings from monogenic forms of obesity. With these new pipelines, the aim to target the cell types involved. CELLEX combined 4 gene expression specificity measures into one score to indicate in which cell type a given gene is specifically expressed. CELLECT quantifies the enrichment of heritability in/near genes specifically expressed in certain cell types. By applying CELLEX and CELLECT to genes prioritized from GWAS for BMI, they identified 26 neuronal cell types across the brain, rather than being restricted to a limited number of tissues.CELLECT depends on other prioritization software, such as S-LDSC, MAGMA, DEPICT, etc. For the current analyses, the authors used S-LDSC. How confident can we be that the genes prioritized are indeed the causal genes and what would the results look like if DEPICT (the authors' own software), or MAGMA were used? At least a justification for using S-LDSC needs to be given.

We used S-LDSC for our primary cell type prioritization results because Finucane et al. reported that S-LDSC is superior to other tissue and cell type prioritization methods (ref. (6), Supplementary Figure 16). Specifically they found that S-LDSC had more power to detect causal annotations than the non-polygenic DEPICT method and that MAGMA suffers from higher false-positive rates compared to S-LDSC (partly due to uncorrected genomic confounding). Consequently we built CELLECT around the S-LDSC framework. To make this more clear, we modified the following sentence in the Results: “Here, we – due to its polygenic nature and well-controlled type I error rate – used CELLECT with S-LDSC as the genetic prioritization model to quantify the effects of cell type ESμ on BMI heritability.”

We do consider MAGMA a fast and efficient method for cell type prioritization and since submission, we have implemented CELLECT-MAGMA as part of the CELLECT workflow (https://github.com/perslab/timshel-bmicelltypes) and added a CELLET-MAGMA documentation and tutorial on how to best run it. We compared our BMI GWAS results obtained with S-LDSC to MAGMA and found that almost all of the S-LDSC prioritized cell types were also prioritized by MAGMA, but MAGMA tended to prioritize more cell types (Figure 3—figure supplement 3b). This overall observation is consistent with the Finucane et al. results. We did not build a version around DEPICT because DEPICT is not fully polygenic in its setup as it is limited to a given number of top associated GWAS loci. Please note that CELLECT leverages existing methods, such as S-LDSC and MAGMA, to prioritize cell types and not genes. Genes should be prioritized using DEPICT, MAGMA, GRAIL or other methods.

The assumption for CELLECT is that genes that have a high expression specificity (e.g. in the brain) are more important than genes that are widely expressed (or not expressed much) across cell types and tissues. How confident can one be that this is a correct assumption (across the board)? More specifically, are genes that are widely expressed or not expressed or expressed in other tissues not important in obesity? For example, LEP is predominantly expressed in adipose tissue, but is known to signal to the brain to influence food intake.

This is a very relevant point. We modelled our work on previous work showing that risk genes tend to be specifically expressed (7). We acknowledge that this assumption and design leads to a dependence on the other cell populations in the dataset. This is an important limitation. We recommend running a “tiered” prioritization strategy for CELLECT, where, for a given complex trait, one starts with analysing body-wide or organ-wide transcriptional atlases and then turns to more tissue-centric datasets. In such a setup we would predict the leptin gene to be specifically expressed in mature adipocytes (because it is almost exclusively expressed in mature adipocytes). Unfortunately, neither the Tabula Muris dataset, nor any of other published multi-organ datasets, contain a mature adipocyte cell population to test this hypothesis. For another example we turned our attention to the high-confidence obesity genes; among the 23 genes, with the exception of Zbtb7b, all genes were detected as being specifically expressed in at least one cell type across the Tabula Muris, Mouse Nervous System and ARC datasets. Furthermore, across these four datasets, 22 of the 23 genes were among the 25% most specifically expressed genes in at least one cell type. We added the following changes to the manuscript:

Results: “Finally, we found that across the Tabula Muris, Mouse Nervous System and ARC datasets, 22 of the 23 high-confidence obesity genes were among the 25% most specifically expressed genes in at least one cell type”.Discussion: “Finally, given the dependence of CELLECT results on other cell types in the given datasets, we, generally, recommend running a “tiered” prioritization strategy for CELLECT, where one preferably starts with analyzing body-wide or organ-wide transcriptional atlases and then turns to more tissue-centric datasets”.Added Figure 5—source data 9 supporting the above results.

While there is indeed growing evidence that not only the hypothalamus plays a key role in obesity, one would expect that well-established obesity genes, tissues and cell types would be identified by new methods.

As noted above, we have now performed additional analyses for the hypothalamus and updated the manuscript.

According to the Tabula Muris Nature paper (2018), it includes cerebellum, cortex, hippocampus and striatum. Can cell types be distinguished by these tissues, in particular given that the hippocampus was the most enriched tissue in Locke et al., one may expect some more specific evidence for this tissue/cell type.

The Tabula Muris study has sampled all mouse tissues at a relatively low resolution. In our work the Tabular Muris analysis is intended to provide us with overall direction regarding the parts of the body on which to focus our attention. For this analysis, we did not use the subtissue annotation, which contains the cerebellum (n=1,317 neuronal cells), cortex (n=56), striatum (n=106) and hippocampus (n=19). To do a more thorough analysis of certain areas in the brain, such as the hippocampus, we used the Mouse Nervous System dataset. In the latter dataset, we identified three hippocampal cell types supporting our previous findings from Locke et al. (8).

It was stated that coding mutations in "syndromic" forms of obesity were chosen for replication of methods. It should be noted that this list includes genes identified for monogenic and syndromic forms of obesity – it seems the term "syndromic" is not used correctly (ie. Include monogenic and syndromic). While not unimportant, it seems that mutations in monogenic forms of early-onset extreme obesity are more relevant for common forms of obesity than syndromic. Therefore, these analyses may be better when stratified.

We would like to thank the reviewer for pointing this out We have now excluded syndromic obesity genes from our analyses to only base our work on genes leading to monogenic forms of obesity, extreme obesity and genes harboring protein-altering variants associated with BMI (all from Turcot et al.(9)). For simplicity, we now refer to that set of 23 genes as “high-confidence obesity genes”. Omitting the genes associated with syndromic obesity and only focusing on the remain 23 high-confidence obesity genes changed the results slightly:

Instead of 15 Mouse Nervous System dataset cell types, we now find that 8 are significant (MEGLU2 and DEGLU5 overlap with the CELLECT analysis), including a glutamatergic neuronal cell population from the ventromedial hypothalamus (HYPEP3).For the Mouse Nervous System cell types, the correlation between the high-confidence obesity gene set enrichment and the CELLECT scores decreased slightly from Pearson‘s rho=0.58 (*P*=1.7×10^-25^) to Pearson’s rho=0.54 (*P*=3.0x10^-21^).Added Figure 5—source data 6 showing the high-confidence obesity geneset enrichment for all single-cell datasets used in this work.We updated the Results and added the above-mentioned additional analysis based on the high-confidence obesity geneset.

Reviewer #2:More than 250 genetic loci have been implicated in human obesity. While others have shown that the GWAS loci are disproportionately expressed in the brain, the key cell types affected is not known. The authors developed a computational pipeline that leverages the growing body of single cell RNA-Seq data to identify specific cell types that are likely to be preferentially impacted by the GWAS variants. The authors developed two computational tools that are released as open-source packages for Python programming languages: CELLEX and CELLECT. Application of CELLEX (Cell type EXpression-specificity) to scRNA-seq data integrates four metrics of expression specificity into a single parameter (Expression Specificity, ES) to identify genes that are preferentially expressed in a particular cell type. CELLECT (Cell type Expression specific integration for Complex Traits) integrates the information from CELLEX with GWAS data to identify cell types that have enriched expression of nearby genes, and thus are likely to contribute to the pathophysiology of obesity. They validated the CELLECT tool by showing that it could identify relevant cell types for 10 different GWAS databases (i.e. neurons in the case of the BMI GWAS). Running this analytical pipeline on 256 from the mouse nervous system scRNA-seq and BMI GWAS databases identified 22 enriched cell types in 8 brain regions. Surprisingly, none of these cell types were hypothalamic, which could be explained by sparse sampling of the large number different hypothalamic cell types in the scRNA-seq datasets. Direct interrogation of 347 hypothalamic cell types identified 4 enriched cell types in the VMH, LHA and POA. If their assumptions and models work as predicted, these tools would significantly advance efforts to uncover cell types and circuits regulating susceptibility to obesity, or any other complex trait of interest. While any computational toolkit has its limitations, several issues must be addressed in order to evaluate the reliability of the data generated with this pipeline.1) The authors need to explain why PVH and ARH neuronal cell types were not identified in their analyses. POMC and MC4R are GWAS loci, and mutations in these genes produce severe obesity in humans and in mouse models. CELLEX/CELLECT implicated a cortical cell type in mediating the effects of mutations in MC4R and the ciliary gene ADYC3. Genetic studies in mice provide strong evidence that disruptions of MC4R or cilia formation in the PVH is sufficient to cause obesity.

This is a very relevant concern; please refer to our above response to the two summarized major concerns.

(1a) To what degree are the analyses impacted by differences in the representation of cell types in brain regions that are very heterogeneous? Are ES scores inflated in brain regions where the cell types are more homogeneous or better annotated? To what extent will this issue be mitigated as more scRNA-seq datasets are published in heterogeneous tissue such as the hypothalamus?

This is a relevant concern which we have tried to address in our responses above.

(1b) The implication of a cortical cell type in mediating influences of MC4R and ADYC3 is novel and unexpected. Demonstration that disruption of MC4R or cilia formation in the cortex would go a long way to allay concerns about the validity of the CELLEX/CELLECT toolkit.

We agree that it is important to follow up on the prioritized cell types and that the MC4R- and that the Adcy3-expressing cortical TEGLU4 cell types could be an interesting candidate. However, this observation (Figure 1 in the Supplementary file 1) was meant as an example and we have now replaced the former sentence in legend of the supplementary figure (“TEGLU4 (located in cortex pyramidal layer 5) is a candidate etiologic cell type mediating the role of MC4R/ADCY3 in obesity”) with “The co-specific expression of Mc4r and Adcy3 may serve as an example on how certain cell types may co-express a core gene (Mc4r in this example) and peripheral genes (Adcy3 in this example)”.

Reviewer #3:The authors report the development of in silico tools that aspire to help identify cell types that participate in body weight regulation by aligning BMI GWAS loci with scRNA-seq datasets spanning an array of tissues. The key assumption for this analysis is that "in order for a disease to manifest in a given tissue or cell type the set of disease-causing genes must be active and expressed in the given tissue or cell type. In other words, the model presupposes that high/increased expression (and not decreased/lack-of expression) of a gene results in disease." The wording states poorly this important assumption. The authors assume that in order for a gene to be important in a specific tissue, it needs to be expressed. It has nothing to do with the change in expression of a certain gene per SNP allele. If the gene is not expressed in a certain cell type, it is unlikely that the presence of one allele or the other influences the disease or trait.

We thank the reviewer for pointing out the confusing formulation. While the formulation is far too wordy, our model actually assumes a linear effect of cell type expression specificity (please refer to Supplementary file 1) and trait heritability. We have now reworded to corresponding sentence to: “Third, one should keep in mind the overall assumption behind our approach, namely that in order for a given gene to confer genetic susceptibility for a given disease it needs to be expressed in the given cell type or tissue, where increasing expression is associated with increasing relevance” (Discussion).

1) In addition to the narrow field of cell state (adult versus development) that Timshel et al. point out, two important limitations/confounds are not highlighted: a) scRNA-seq depth of sequencing limits the detection of lowly expressed upstream signaling components critical to body weight regulation and b) gene prioritization is based on a variety of criteria that vary from one GWAS study to the other. Calling the wrong gene would affect the input in the present analysis.

We have addressed the highly relevant concern related to sequencing depth in our above response to the two summarized major concerns. Regarding point (b), our analysis is not based on a particular set of genes prioritized in the source GWAS and hence not dependent on the gene prioritization algorithm that may have been applied in conjunction with the original GWAS study.

2) Given all these limitations, a proof of principle example should be tested. Mc4r is the perfect candidate, alas the absence of scRNA-seq data from highly relevant cell types including the PVH. The fact that hypothalamic BMI GWAS enrichment is low runs against the validation of the tool. So does the leptin receptor result, possibly indicative of the limitation of scRNA-seq transcript capture.

As described above, we agree that PVH Mc4r^+^ cells would have constituted a good proof of principle example. Furthermore, we realized that Figure 6 needs further explanation. Figure 6 only shows Mouse Nervous System cell types, in the added Figure 5B, we show the expression of the leptin receptor across hypothalamic cell types. Lepr is both expressed and specifically expressed in two relevant neuronal cell populations namely Agrp- and Trh/Cxcl12-expressing cells (two known leptin-sensing cell populations (10)). We updated the legend of Figure 6: “The plot shows gene ES_μ_ (y-axis) for each cell type (x-axis, ordered by increasing values of expression specificity, ES_μ_) with BMI-prioritized cell types from the Mouse Nervous System dataset highlighted”.

3) The authors state that CELLEX is currently not set up for adjustment of unwanted sources of variation such as batch effects. The authors should clarify the magnitude of variation that is tolerated without affecting the ES metrics.

It is correct that CELLEX does not explicitly adjust for batch effects. Adjusting single-cell RNA-seq data is a non-trivial task and major integration tools like Seurat (11), Liger (12) and Harmony (13) return reduced space representations of the integrated datasets, which are non-linear transformation of the original expression values that are incompatible with the expression specific measures CELLEX relies on. However, because CELLEX is operating on averages of gene expression measures, it is relatively robust to batch effects. The analysis of the hypothalamus datasets provides an illustrative example. In that analysis, we separately ran each hypothalamus dataset through CELLEX. Despite the fact that these datasets were created in different labs with different techniques, clustering of the cell populations from the six datasets shows that cell populations predominantly cluster by cell type (e.g. neurons clustered distinctly from glial cells, and microglia and brain perivascular macrophages clustered distinctly from other glial cell populations) rather than by study (Figure 1—figure supplement 2). These results confirm that CELLEX is relatively robust to batch effects. We have now added the following sentence to the Materials and methods: “Noteworthy, clustering of the 347 hypothalamic cell populations based on their ES_μ_ values resulted in clusters predominantly separating by cell type rather than by study or single-cell technique, indicating that CELLEX is relatively robust to batch effects”.

4) Furthermore, given this limitation of CELLEX, how were the different datasets from the hypothalamus combined? Was each dataset run separately through CELLEX or were the 347 cell types combined to a CELLEX input? Does CELLEX take count data as input or does it work with other measures such as TPM?

We thank the reviewer for making us aware of that missing description on how we analyzed the hypothalamic datasets. To best account for batch effects and the above “relativity” issue, each hypothalamus dataset was run separately through CELLEX and CELLECT. Bonferroni correction was applied on all 347 hypothalamic cell types to obtain the final number of significant cell populations across the six hypothalamic datasets. This decision was motivated by the heterogeneity of the hypothalamus and the fact that most of the six studies have focused on relatively distinct parts of the hypothalamus resulting in little overlap between the studies. In the Materials and methods we mentioned that the hypothalamic datasets were “analyzed separately in CELLEX and CELLECT”.

CELLEX can take count data as input as well as transcripts per million (TPM)-normalized data. The Tabula Muris (SMART-seq2) analyses were based on TPM-normalized data. We have updated the GitHub documentation to make this more clear to CELLEX users, and updated the Materials and methods: “CELLEX can take count data as well as transcripts per million-normalized data as input”.

5) The use of the WGCNA method does not seem to serve a purpose. If the authors need to undertake network analysis, they should utilize methods developed for scRNA-seq or completely omit this from the manuscript.

We kindly disagree, WGCNA has successfully been used to analyze droplet based single-cell RNA-seq data in several studies, including refs.(14). We too have had good experiences with WGCNA across our single-cell analyses, which is why we used it in this paper. We do acknowledge that it comes with limitations when applied on cells from a single cell type and discuss these in Supplementary file 3.

[Editors' note: further revisions were suggested prior to acceptance, as described below.]

The reviewers agreed that the paper has improved substantially and the authors have addressed their concerns. Nevertheless, there are some remaining concerns of which we'd like you to address the following [no need to address the individual reviewer comments at this point];– Both reviewer 1 and 2 would like you to put your results/observations into context; what does one do next with the "identified" genes and tissues.

We agree that this indeed would be useful and have now added the following section to the Discussion:

“Strategies for follow-up

Having identified GWAS-enriched cell populations only marks the start of the journey towards understanding how genetic variants render us susceptible to obesity. […] Given that CELLEX provides marker genes specifically marking the focal cell population and that all enriched cell populations were of neuronal origin, transgenic animal model techniques such as designer receptors exclusively activated by designer drugs (DREADD)-based chemogenetic tools for activation or inhibition of neurons, transgenic techniques for cell ablation, and fiber photometry techniques for real-time monitoring the impact of relevant physiological environments or pharmacological treatments on the focal cell type, are well-positioned to provide relevant insights into the role of the given cell type in the control of energy homeostasis.”

– Reviewer 2 would like to make sure your tool does readily identify the genes driving the link a specific tissue/cell type.

We have now added scripts to allow users to more readily identify candidate genes driving prioritization of a given cell type. We have called the functionality CELLECT-GENES and it provides, for each GWAS and single-cell RNA-seq dataset, the set of genes with the largest association signals and highest expression specificities across enriched cell types. The CELLECT-GENES scripts can be found in the CELLEC repository (github.com/perslab/CELLECT) and a tutorial on how to run it can be found along with the other CELLEX and CELLECT tutorials (github.com/perslab/CELLECT/wiki/CELLECT-GENES-Tutorial).

**References**

1) Campbell et al., “A Molecular Census of Arcuate Hypothalamus and Median Eminence Cell Types.”

2) An et al., “TrkB-Expressing Paraventricular Hypothalamic Neurons Suppress Appetite through Multiple Neurocircuits.”

3) Khera et al., “Polygenic Prediction of Weight and Obesity Trajectories from Birth to Adulthood.”

4) van der Klaauw et al., “Human Semaphorin 3 Variants Link Melanocortin Circuit Development and Energy Balance.”

5) Akiyama et al., “Genome-Wide Association Study Identifies 112 New Loci for Body Mass Index in the Japanese Population”; Locke et al., “Genetic Studies of Body Mass Index Yield New Insights for Obesity Biology.”

6) Finucane et al., “Heritability Enrichment of Specifically Expressed Genes Identifies Disease-Relevant Tissues and Cell Types.”

7) Smillie et al., “Intra- and Inter-Cellular Rewiring of the Human Colon during Ulcerative Colitis.”

8) Locke et al., “Genetic Studies of Body Mass Index Yield New Insights for Obesity Biology.”

9) Turcot et al., “Protein-Altering Variants Associated with Body Mass Index Implicate Pathways That Control Energy Intake and Expenditure in Obesity.”

10) Campbell et al., “A Molecular Census of Arcuate Hypothalamus and Median Eminence Cell Types.”

11) Stuart et al., “Comprehensive Integration of Single-Cell Data.”

12) Welch et al., “Single-Cell Multi-Omic Integration Compares and Contrasts Features of Brain Cell Identity.”

13) Korsunsky et al., “Fast, Sensitive and Accurate Integration of Single-Cell Data with Harmony.”

14) Nowakowski et al., “Spatiotemporal Gene Expression Trajectories Reveal Developmental Hierarchies of the Human Cortex”; Luo et al., “Single-Cell Transcriptome Analyses Reveal Signals to Activate Dormant Neural Stem Cells”; Skinnider, Squair, and Foster, “Evaluating Measures of Association for Single-Cell Transcriptomics.”